# Know Your Self-supervised Learning: A Survey on Image-based Generative and Discriminative Training

**Utku Ozbulak**[1,2]                     *utku.ozbulak@ghent.ac.kr*
**Hyun Jung Lee**[1,2]                   *hyunjung.lee@ghent.ac.kr*
**Beril Boga**[3]                            *beril.boga@bshg.com*
**Esla Timothy Anzaku**[1,2]        *eslatimothy.anzaku@ghent.ac.kr*
**Homin Park**[1,2]                       *homin.park@ghent.ac.kr*
**Arnout Van Messem**[4]            *arnout.vanmessem@uliege.be*
**Wesley De Neve**[1,2]                *wesley.deneve@ghent.ac.kr*
**Joris Vankerschaver**[1,2]         *joris.vankerschaver@ghent.ac.kr*

[1] *Ghent University, Belgium*
[2] *Ghent University Global Campus, South Korea*
[3] *BSH Hausgeräte GmbH, Germany*
[4] *University of Liège, Belgium*

**Reviewed on OpenReview:** *https://openreview.net/forum?id=Ma25S4ludQ*

## Abstract

Although supervised learning has been highly successful in improving the state-of-the-art in the domain of image-based computer vision in the past, the margin of improvement has diminished significantly in recent years, indicating that a plateau is in sight. Meanwhile, the use of self-supervised learning (SSL) for the purpose of natural language processing (NLP) has seen tremendous successes during the past couple of years, with this new learning paradigm yielding powerful language models. Inspired by the excellent results obtained in the field of NLP, self-supervised methods that rely on clustering, contrastive learning, distillation, and information-maximization, which all fall under the banner of discriminative SSL, have experienced a swift uptake in the area of computer vision. Shortly afterwards, generative SSL frameworks that are mostly based on masked image modeling, complemented and surpassed the results obtained with discriminative SSL. Consequently, within a span of three years, over 100 unique general-purpose frameworks for generative and discriminative SSL, with a focus on imaging, were proposed. In this survey, we review a plethora of research efforts conducted on image-oriented SSL, providing a historic view and paying attention to best practices as well as useful software packages. While doing so, we discuss pretext tasks for image-based SSL, as well as techniques that are commonly used in image-based SSL. Lastly, to aid researchers who aim at contributing to image-focused SSL, we outline a number of promising research directions.

## 1 Introduction

The remarkable feature extraction capabilities of deep neural networks (DNNs) have enabled their effective utilization in numerous visual tasks. Although the core building blocks that are in common use today were already proposed two decades ago (LeCun et al., 1998), DNNs only became the go-to models after the introduction of AlexNet (Krizhevsky et al., 2012), a DNN architecture that was able to obtain exceptional results for the ImageNet Large Scale Visual Recognition Challenge (Russakovsky et al., 2015) that took place in 2012, by leveraging vast amounts of computational resources (at that time) and large amounts of labeled data. Since then, the availability of standardized datasets in the image domain such as MNIST (LeCun et al., 1998), CIFAR (Krizhevsky & Hinton, 2009), SVHN (Netzer et al., 2011), COCO (Lin et al., 2014),

and ImageNet enabled standardized experimentation, with these datasets acting as catalysts for major advancements in the area of supervised learning. Starting with AlexNet, the classification accuracy of DNNs on ImageNet improved year after year thanks to better and novel architectural designs (e.g., VGG (Simonyan & Zisserman, 2015), ResNet (He et al., 2016), InceptionNet (Szegedy et al., 2015; 2016), ViT (Dosovitskiy et al., 2020)), augmentation techniques, optimizers, and activation functions, as well as methods for smoother training (Loshchilov & Hutter, 2017; Yun et al., 2019; Ioffe & Szegedy, 2015; Kingma & Ba, 2014; Clevert et al., 2015).

Unfortunately, not all datasets come with an abundance of labeled training data. In order to overcome this hurdle and to facilitate the application of DNNs to smaller datasets, transfer learning was introduced and soon became the dominant method to transfer knowledge across image datasets (Tan et al., 2018). Although transfer learning enables the usage of DNNs for smaller datasets thanks to features extracted from larger datasets, models trained in this way are known to be brittle and sensitive to small changes in the data (Jain et al., 2022) due to the use of supervised pre-training. Furthermore, shortcomings of supervised learning also became apparent when improvements obtained with these methods came to a halt in recent years (see Figure 1 for top-1 accuracy on ImageNet), thus calling for research efforts that go beyond the use of supervised learning (Zisserman, 2018). In order to overcome the limitations of supervised learning, countless studies investigated the line of unsupervised learning, which aims at enabling robust feature extraction through the training of models without label information (Celebi & Aydin, 2016). Unfortunately, results obtained by these methods on image datasets fell short until recently (Noroozi & Favaro, 2016; Pathak et al., 2016), while the use of self-supervised methods in the field of natural language processing (NLP) achieved state-of-the-art results, compared to supervised learning techniques (Devlin et al., 2018; Radford et al., 2019).

As mentioned above, the field of NLP enjoyed the success of self-supervised models over supervised ones earlier than the field of computer vision, with models such as `BERT`, `GPT`, and their variants achieving state-of-the art results (Devlin et al., 2018; Radford et al., 2019; Brown et al., 2020). One reason which explains the success of SSL in NLP is the abundance of unlabeled text data, such as books, online websites, and blogs (Chen et al., 2017; Hamilton et al., 2017), which prompted researchers to investigate SSL over supervised training. Another reason that explains their success, as discussed by He et al. (2020), is the fundamental difference between the signal space of NLP and the signal space of computer vision, given that language data are discrete and structured (i.e., words), whereas image data are high dimensional, continuous, and unstructured. Nevertheless, we can state that the success of SSL in the field of NLP prompted the computer vision community to put more investigative efforts into this learning paradigm.

In order to alleviate issues regarding label requirements, as well as to enable robust feature extraction, self-supervised learning in computer vision emerged as a method for extracting robust features from unlabeled data using the properties of images themselves (He et al., 2020; Chen et al., 2020b). The idea behind SSL is straightforward: devise an experimental setting in which the task that provides the supervisory signal can be solved without human annotation and then train DNNs to solve it.

Note that the description provided above for SSL also covers a number of additional approaches including autoencoders (Gogna & Majumdar, 2016), generative models, and clustering-based methods that leverage self-labeling (Caron et al., 2018), and that these approaches also fall into the category of unsupervised learning (since human annotation is not necessary). Furthermore, most of the training routines described in this manuscript also use the term "self-supervised learning" interchangeably with "representation learning" when supervision is provided by the data, while representation learning is described by Bengio et al. (2013) as "learning representations of the data that make it easier to extract useful information when building classifiers or other predictors", irrespective of the supervisory nature of the learning methodology. So, how did "self-supervision" become such a popular term in recent years?

**Resurgence of the term "self-supervised learning" in computer vision** – Beyond a number of niche use cases such as image colorization (Larsson et al., 2017), image inpainting (Yang et al., 2017), and puzzle-solvers (Trinh et al., 2019) that explicitly use self-supervision, the term "self-supervised learning" was previously not employed to describe many techniques. Furthermore, compared to other learning paradigms, the use of SSL was not popular until recently (see Figure 1). In fact, research efforts that are now considered to be pioneers in SSL and that are used for SSL benchmarking, such as `Deep Cluster` (Caron et al., 2018),

(a) ImageNet top-1 accuracy

(b) Interest over time for different learning paradigms

Figure 1: (a) ImageNet top-1 accuracy for DNNs proposed between 2012 - 2022 and (b) interest over time for three popular learning paradigms between 2004 - 2022, as measured with Google Trends.

`InstDist` (Wu et al., 2018b), `CPC` (Oord et al., 2018), and `Local Aggregation` (Zhuang et al., 2019), were published as unsupervised training methods, distancing themselves from SSL.

The resurgence of interest in self-supervision and the re-branding of corresponding methodologies can be attributed to the popularization of the term by both authoritative researchers and tech giants in the field between 2018 and 2020 (Zisserman, 2018; Efros, 2019; Bachman, 2019; LeCun & Misra, 2020; Chen, 2020; Howard, 2020). The reason for this re-branding is straightforward: most of the tasks discussed above that fell under the banner of unsupervised learning were deemed misleading, since the training was not completely unsupervised. Instead, the supervision was provided by the data itself, without explicit human labeling (Zisserman, 2018; LeCun, 2019). As a result of this re-branding, while most papers published before 2020 use unsupervised learning to describe their work, those that are published after 2020 use the description self-supervised learning, hence the conflict between the use of the two terms.

An interesting moment in this timeline, and the one that furthered the popularity of the term SSL, is the revision by Yann LeCun of his now-famous cake analogy from NeurIPS-16, during a talk he gave at ISSCC-19 and later at AAAI-20 (LeCun, 2020): "If intelligence is a cake, the bulk of the cake is ~~unsupervised~~ *self-supervised* learning, the icing on the cake is supervised learning, and the cherry on the cake is reinforcement learning" (LeCun, 2016).

In summary, we can say that self-supervised learning refers to a recently popularized learning paradigm, encompassing predictive tasks where the supervisory signal is provided by the data, without relying on the explicit use of human labels.

**Generative and discriminative SSL** – In general, self-supervision approaches can be grouped into two categories: generative and discriminative (Doersch et al., 2015). In generative self-supervision, the task is to build appropriate distributions over a collection of data while operating in the pixel space. A common criticism of generative self-supervision is that it is computationally expensive, does not work well with high-resolution images, and that it may be superfluous for representation learning (Chen et al., 2020b; Grill et al., 2020). Typical models relying on this kind of self-supervision are autoencoders (AEs) and generative adversarial networks (GANs) (Kingma & Welling, 2013; Vincent et al., 2008; Goodfellow et al., 2020). It should be noted that although both AEs and GANs are categorized as "generative" models, they achieve self-supervision in different and distinct ways.

In contrast to generative SSL, in discriminative self-supervision, the task is to learn good representations of the data in order to perform a specified pretext task (which we will explain shortly) that does not require a human annotation effort (Doersch et al., 2015). Discriminative self-supervision is similar to supervised learning in the sense that the objective function is often a scoring function that evaluates the discriminative power of learned representations. Most of the SSL frameworks we will cover in this manuscript refer to the works of Becker & Hinton (1992) and Bromley et al. (1993) as the earliest research efforts that use discriminative self-supervision in the form it is used nowadays, with the above research efforts investigating representation alignments across different inputs.

**Purpose of this survey** – Thanks to the excellent results obtained by SSL in computer vision, numerous SSL frameworks were proposed within the span of a couple of years. Although most of these frameworks are often specialized in nature, addressing a select number of tasks (such as depth estimation, face recognition, remote sensing, and pose estimation), we could trace their origin to roughly 100 general-purpose SSL frameworks that are applicable to images. Even though several in-depth surveys are available on the topic of image-

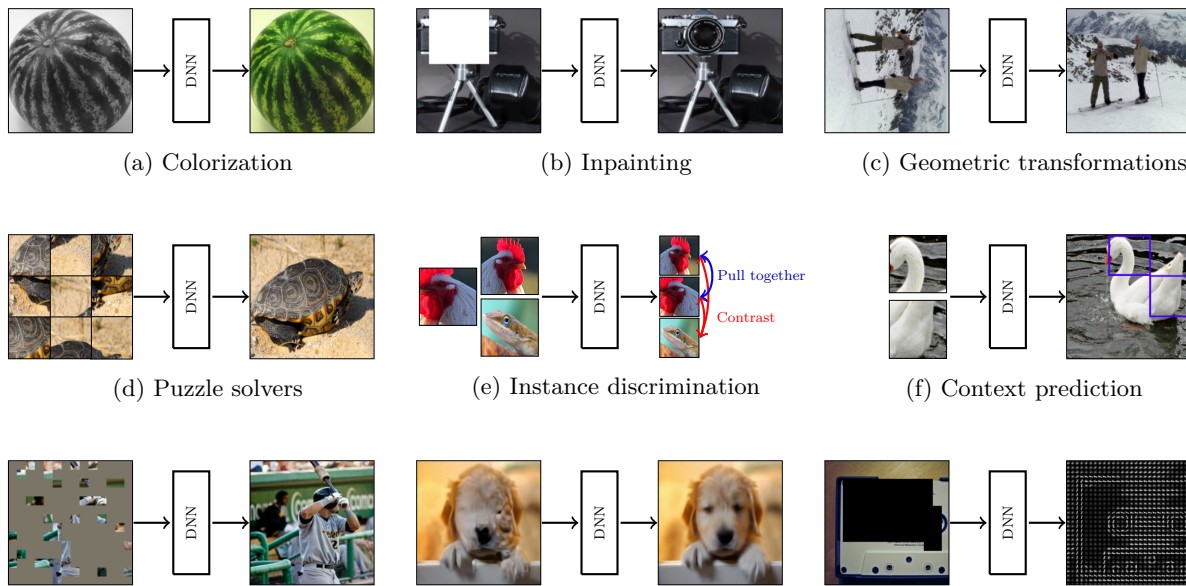

Figure 2: Illustrations of various image-based pretext tasks for self-supervised learning.

based contrastive SSL (Albelwi, 2022; Khan et al., 2022), due to the fast-paced nature of research in SSL, they do not cover recent non-contrastive SSL methods that transformed the field. As such, a major goal of this survey is to cover the aforementioned image-oriented frameworks for generative and discriminative SSL which benefited from a tremendous research and development efforts in recent years, hereby presenting a concise and aggregate work to readers who take an interest in this field.

In Section 2, we describe popular pretext tasks for self-supervision, subsequently detailing a number of relevant technical concepts that are commonly used in Section 3. Diving deeper into SSL as it is used nowadays for image-related tasks, in Section 4, we cover recently proposed SSL frameworks for image-based training in a chronological order and discuss methods of evaluation in Section 5. In Section 6, we cover relevant libraries, repositories, and publicly available implementations that aim at assisting researchers. Finally, in Section 7, we review a number of shortcomings of SSL, identify open problems, and conclude our survey.

## 2 Pretext tasks for self-supervised learning

The image domain allows a number of unique pretext tasks that enable self-supervision. Below we describe the most popular ones and illustrate them in Figure 2.

**Image colorization** – Automated colorization of grayscale images is a line of research that was investigated even before the widespread usage of DNNs (Luan et al., 2007; Charpiat et al., 2008). However, the availability of large-scale colored datasets such as ImageNet, combined with the versatility of DNNs, further strengthened the interest in high-quality image colorization, especially for the purpose of coloring historical pictures. In parallel to research efforts that aimed at increasing the quality of colorization, such as (Cheng et al., 2015; Iizuka et al., 2016), the idea of using image colorization as a pretext task for representation learning was also investigated (Larsson et al., 2017; 2016; Zhang et al., 2016). Although this task alone was revealed to be too simple to force DNNs to learn complex representations (Caron et al., 2020), colorization is still used in tandem with other tasks to boost the effectiveness of SSL models.

**Inpainting** – The task of predicting a missing part of an image is referred to as image inpainting (Bertalmio et al., 2000). With the widespread usage of DNNs, inpainting problems also found numerous solutions (Yang et al., 2017; Yu et al., 2019). One such solution, and the one that allows for the use of SSL, is proposed

by Pathak et al. (2016), leveraging context encoders that aim at inpainting large parts of images that are missing, forcing models to learn the image context.

**Geometric transformations** – Inspired by research efforts that bring together geometric transformations and neural networks (Kanazawa et al., 2016; Rocco et al., 2017), and taking advantage of image-based datasets that almost always contain upright images, Gidaris et al. (2018) proposed the idea of predicting image rotations as a method of self-supervision. Following the success of this method, other types of geometric transformations were proposed by Novotny et al. (2018); Zhang et al. (2019); Chen et al. (2019).

**Puzzle solvers** – A unique image-based task that can be formulated in a SSL setting is solving a jigsaw puzzle (Noroozi & Favaro, 2016), where the goal is to correctly predict the relative location of nine puzzle pieces. This unusual pretext task, as well as a number of derivations, is employed in support of a variety of tasks, including domain generalization (Carlucci et al., 2019), generation of image embeddings (Trinh et al., 2019), image retrieval (Pang et al., 2020), and auxiliary learning (Li et al., 2021b).

**Instance discrimination** – Given differently augmented views (i.e., instances) originating from one image, instance discrimination refers to the idea of recognizing these views as originating from the same image, while discriminating any other image with a different origin (Wu et al., 2018b). Different from the previously described pretext tasks which achieve representation learning as a by-product of the optimization objective, instance discrimination optimizes for representation learning by directly matching the representations of similar images while contrasting the representations of dissimilar ones. In this context, images that are contrasted to similar ones are called negative samples (e.g., the gecko image in Figure 2e). The main idea behind representation matching between similar images and contrasting different images is to help DNNs learn representations that are invariant to commonly used image transformations, since most of these transformations do not alter the visual semantics (Misra & Maaten, 2020). The origins of this approach can be traced back to the research efforts presented in Hadsell et al. (2006), Sohn & Lee (2012), and Hui (2013).

**Masked image modeling** – The adaptation of masked language modeling in NLP to computer vision as a new pretext task for self-supervised training was a groundbreaking discovery in generative SSL (Chen et al., 2020a; Bao et al., 2021). This technique is referred to as masked image modeling (MIM) (Bao et al., 2021). The idea behind MIM is simple: divide an image into a collection of equal-sized patches, mask some of the patches, and task the model with generating their corresponding pattern. As we will discuss in later parts of this paper, while the usage of MIM has popularized generative SSL, this pretext task can be thought of as a variant of image inpainting. The primary difference between MIM and the inpainting method proposed in the work of Pathak et al. (2016) is that MIM uses non-overlapping patches of equal size. After the rise in popularity of MIM (which is often used in conjunction with vision transformers (ViT) (Dosovitskiy et al., 2020)), a number of its variants emerged, with corrupted image modeling (Fang et al., 2023) (see Figure 2h) and masked feature prediction (Wei et al., 2022) (see Figure 2i) the two most prominent ones.

**Others** – Apart from the mainstream pretext tasks described above, there are a number of unique tasks that do not fit into one of the above categories such as: the split-brain approach which tries to predict a subset of image channels from other channels (Zhang et al., 2017), a feature consistency method involving synthetic images (Ren & Lee, 2018), context prediction (Doersch et al., 2015), adversarial feature learning (Donahue et al., 2016; Donahue & Simonyan, 2019), exemplar networks (Dosovitskiy et al., 2014), and object counting (Noroozi et al., 2017).

**Effectiveness of pretext tasks** – Given the abundance of pretext tasks for self-supervision, which of these tasks enable DNNs to learn the most useful representations? Although there is no clear answer to this question, ever since the works of Dosovitskiy et al. (2014), Wu et al. (2018b), and Oord et al. (2018), instance discrimination was established as the dominant pretext task for image-based discriminative SSL, thanks to the superb results achieved using this type of self-supervision (He et al., 2020; Grill et al., 2020; Chen et al., 2020b). On the other hand, MIM has been recognized as a tremendously powerful pretext task that enables generative SSL to reach and even surpass the results obtained with instance discrimination. (Dosovitskiy et al., 2020; Bao et al., 2021; He et al., 2022).

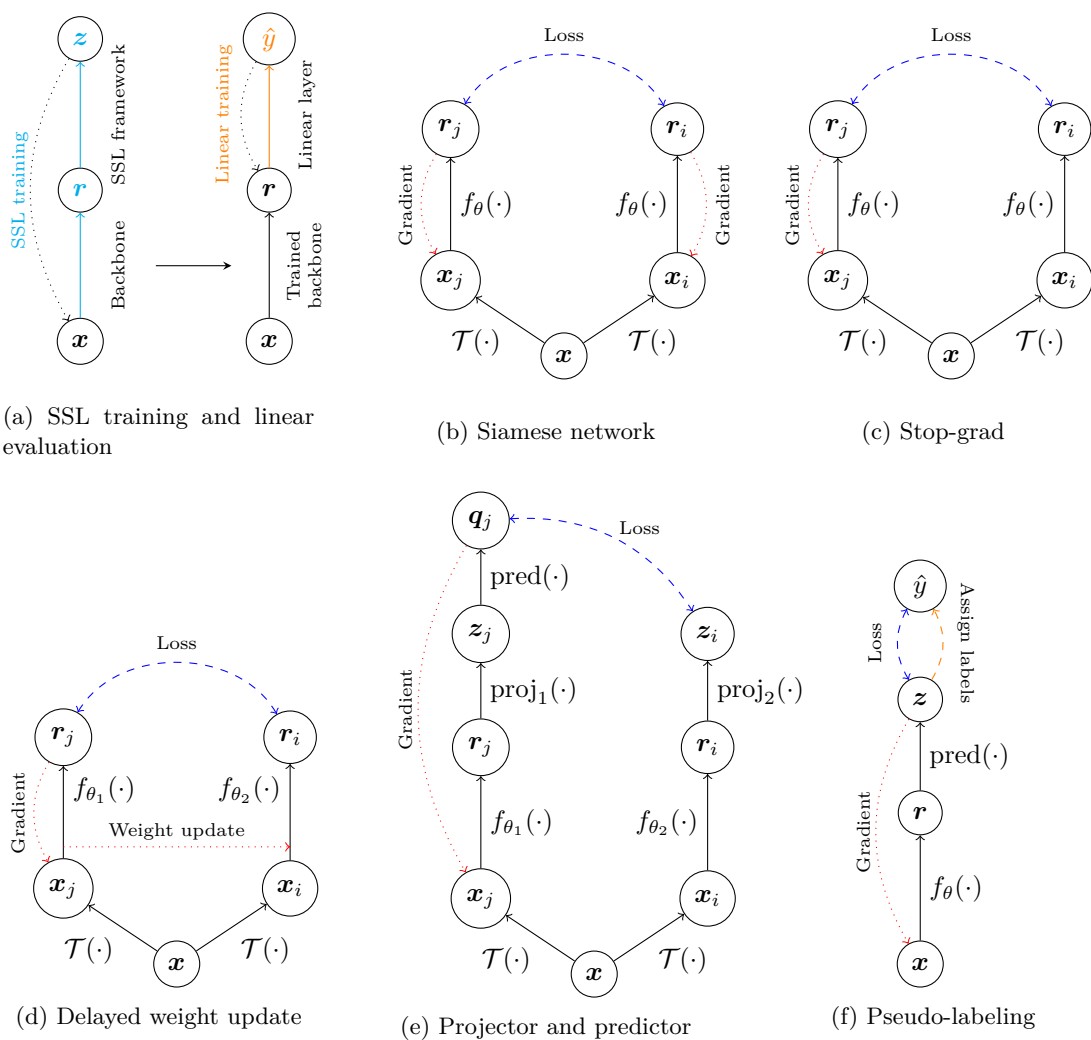

Figure 3: Illustrations of some of the important concepts related to SSL described in Section 3.

# 3 Important concepts in self-supervised learning

In this section, we briefly describe a number of commonly used concepts that are relevant to the forthcoming SSL frameworks. Although these concepts were key elements of early individual SSL frameworks, newer frameworks make use of a mixture of them.

**Notation** – For clarity, we briefly detail the notation used to describe several core SSL concepts. Given an image $\boldsymbol{x} \in \mathbb{R}^p$ and its categorical association $\boldsymbol{y} \in \mathbb{R}^M$ sampled from a dataset $(\boldsymbol{x}, \boldsymbol{y}) \sim \mathcal{D}$, with $y_c = 1$ and $y_m = 0, \forall m \in \{0, \dots, M\} \backslash \{c\}$, let $f_\theta(\cdot)$ be an encoder (i.e., a feature extractor) that maps an image augmented with a stochastic augmentation function $\mathcal{T}(\cdot)$ to a set of features $\boldsymbol{r} \in \mathbb{R}^k$ using a neural network with parameters $\theta$. These features can then be mapped onto a set of projections $\boldsymbol{z}$ and predictions $\boldsymbol{q}$ using the proj($\cdot$) and pred($\cdot$) functions, respectively. In this context, projectors and predictors are simply multi-layer perceptrons (MLP).

**Backbone network** – In the context of SSL, the term "backbone" refers to the feature extractor(s) (i.e., $f_\theta(\cdot)$) that are trained with SSL frameworks. Typically, a backbone network is a task-agnostic DNN (e.g., a ResNet-50 without the final fully connected layer). The majority of the frameworks we will cover use either a variant of ResNet (e.g., vanilla ResNet-50, ResNext, or Wide ResNet) or, very recently, vision transformers as the backbone.

**SSL training and evaluation** – In traditional supervised learning, the feature extractor (e.g., convolutional layers) and the predictor (e.g., linear layers that map features to classes) are trained at the same time. However, SSL is only concerned with the training of the feature extractor. After the SSL training is complete, the linear layer that maps the features to classes is trained separately.

In Figure 3a, we provide a simplified illustration of (left) SSL training and (right) linear evaluation. SSL frameworks are placed on top of backbone networks and are trained in conjunction with the backbone. After the SSL training is complete, the framework is discarded and only the trained backbone is used. Note that this backbone is merely a feature extractor. Then, depending on the problem at hand, a new layer that maps features to classes is initialized and trained. It is crucial to keep in mind that the SSL training is only concerned with the quality of features obtained from the feature extractor. As such, the majority (if not the entirety) of the forthcoming concepts as well as frameworks tackle feature extractor training. Nevertheless, for the sake of completeness, in Section 5, we will also describe evaluation methods.

**Vision transformers** – Vision transformers represent a novel deep learning paradigm that leverages the transformer architecture developed initially for NLP and applies it to image classification tasks.

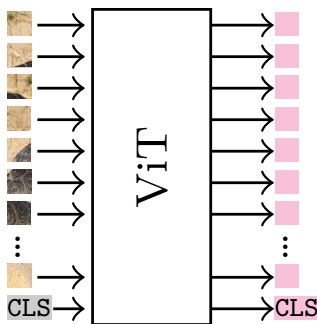

Figure 4: An illustration of ViT input-output relations.

ViT adopts a preprocessing step that involves partitioning the input image into non-overlapping patches, which are linearly embedded to create a sequence of tokens. The transformer encoder is then applied to these tokens, with the self-attention mechanism allowing the model to selectively focus on different patches and learn intricate correlation structures among them.

An essential element of the ViT architecture is the `[CLS]` token, which is prepended to the input and subsequently leveraged for downstream classification tasks. However, in addition to the `[CLS]` token, ViTs also generate patch representation tokens that encapsulate information about the corresponding patch and its relationship with other patches, based on the attention mechanism (see Figure 4). These representation tokens can be utilized for various MIM-based self-supervised tasks, which are relevant for generative SSL frameworks.

**Siamese networks** – A form of dual-backbone networks called Siamese networks (Bromley et al., 1993) consisting of two identical neural networks that share the same set of weights are popular architectures for SSL (see Figure 3b). Although this type of networks was useful in solving a variety of problems (Chopra et al., 2005; Bertinetto et al., 2016; Chicco, 2021), in the context of SSL, they are mostly employed to achieve consistency between representations when, for example, two instances of the same image are provided.

Apart from Siamese networks, a majority of SSL frameworks use dual backbones that may not share weights due to recently discovered beneficial properties. In such cases, the weights of one model are updated via backpropagation, while the weights of the other model can be updated using a variety of techniques which we discuss next.

**Stop-grad** – Siamese networks generally propagate errors from both branches after the loss calculation. As illustrated in Figure 3c, the term "stop-grad" refers to stopping the gradient flow from one branch of a dual-backbone network, while allowing this gradient flow to alter the weights of the other branch (Chen & He, 2021).

**Delayed weight updates** – Assume a Siamese-like dual-backbone network where one branch is called the teacher and the other one the student. However, different than the Siamese architecture, weights of these models are not shared. In this scenario, delayed-weight updates refer to the idea of propagating the error through only one branch via backpropagation and updating the trainable parameters of the other branch via a predetermined rule (see Figure 3d). Popular implementations of this operation are *Mean Teacher* (Tarvainen & Valpola, 2017), *momentum encoding* (He et al., 2020), and *exponential moving average* (Grill et al., 2020).

**Projection and prediction MLPs** – The usage of multi-layer perceptrons in the form of projection and prediction heads following a feature extractor (e.g., a dual backbone) is acknowledged as a powerful technique that greatly improves the effectiveness of SSL methods (Chen et al., 2020d). We visualize this technique in

Figure 3e, as implemented in `BYOL` framework (Grill et al., 2020). Note that this visualization illustrates an asymmetric architecture but the asymmetry is not a necessity for projection/prediction MLPs.

**Negative samples** – The InfoNCE loss (discussed in depth in Section 3.1) aims at maximizing the similarity between representations of two augmentations of the same image, while minimizing the same metric across different images. In such cases, the "different" images are referred to as *negative samples* (Chen et al., 2020b). This concept, which has been the focus of many research efforts (which we will discuss later on), will be particularly relevant for contrastive SSL (He et al., 2020).

**Memory bank** – Given a set of $n$ images, $\mathbf{x} = [\boldsymbol{x}_1, \ldots, \boldsymbol{x}_n]$, a memory bank refers to the simple idea of storing the corresponding image representations, as computed with $f_\theta(\mathbf{x}) = [\boldsymbol{r}_1, \ldots, \boldsymbol{r}_n]$, and to subsequently using this memory bank for various tasks (for example, to use the obtained image representations as negative samples in InfoNCE) (Wu et al., 2018a; He et al., 2020).

**Pseudo-labeling** – A number of SSL methods discussed below employ pseudo-labeling strategies to enable self-supervision (Caron et al., 2018; Asano et al., 2019). Such approaches can be visualized as shown in Figure 3f, where a label is assigned to an image based on its feature representation (through the use of, for example, K-means clustering) and where that label is then used to calculate a loss.

## 3.1 Loss functions to train SSL frameworks

The forthcoming SSL frameworks utilize a wide range of loss functions to enable self-supervised training. Although the usage of these loss functions is often specific to certain frameworks, in this section, we will cover the most prominent losses that see common use across different frameworks.

**Cross-entropy loss** – Cross-entropy loss (CE) is a commonly used loss function in classification tasks which measures the difference between the predicted probabilities and the true probabilities of a categorical variable. Given a prediction $\hat{\boldsymbol{y}}$ for a $C$-class classification problem, CE for the class $t$ is calculated as follows:

$$\mathcal{L}_{\text{CE}}(\hat{\boldsymbol{y}}, t) = -\log \frac{\exp(\hat{\boldsymbol{y}}_t)}{\sum_{c=0}^{C} \exp(\hat{\boldsymbol{y}}_c)}.$$

In clustering-based SSL, CE and its variants are mainly used with the target label $t$ being assigned via a self-labeling mechanism such as k-means clustering (Caron et al., 2018; Asano et al., 2019; Qian et al., 2022). More recently, distillation-based SSL frameworks also make use of CE where the output of the student network is matched to that of the teacher (Caron et al., 2021; Gidaris et al., 2021; Li et al., 2021a).

**Cosine similarity** – Cosine similarity measures the similarity between two non-zero vectors in a high-dimensional space, formalized as a dot product between $\ell_2$ normalized vectors $\boldsymbol{v}_1$ and $\boldsymbol{v}_2$ as follows:

$$\text{sim}(\boldsymbol{v}_1, \boldsymbol{v}_2) = \frac{\boldsymbol{v}_1 \cdot \boldsymbol{v}_2}{\|\boldsymbol{v}_1\|\|\boldsymbol{v}_2\|}.$$

In the context of SSL, cosine similarity is often employed in combination with noise-contrastive estimation (NCE) for contrastive-learning-based discriminative SSL frameworks. It is also employed by a number of prominent distillation networks to quantify representation similarity (Grill et al., 2020; Chen & He, 2021). Given an image $\boldsymbol{x}$ and two views $\boldsymbol{x}_{\{1,2\}} \sim \mathcal{T}(\boldsymbol{x})$ obtained with an augmentation $\mathcal{T}$, let $\boldsymbol{z}_{\{1,2\}}$ and $\boldsymbol{q}_{\{1,2\}}$ be the outputs of the projection and prediction layers, respectively, obtained by using a Siamese-like backbone similar to the one depicted in Figure 5. `SimSiam`, for example, then employs negative symmetric cosine similarity between projections and predictions defined as $-\frac{1}{2}\text{sim}(\boldsymbol{q}_1, \texttt{stop-grad}(\boldsymbol{z}_2)) - \frac{1}{2}\text{sim}(\boldsymbol{q}_2, \texttt{stop-grad}(\boldsymbol{z}_1))$ with `stop-grad`($\cdot$) referring to the stop-grad operation described above (Chen & He, 2021).

**Noise-contrastive estimation** – A contrastive loss is a loss that has a low value when the two input images are similar and a large value when they are dissimilar (Chopra et al., 2005; Hadsell et al., 2006). A fundamental loss that enables contrastive training for image-based SSL is InfoNCE (Sohn, 2016; Oord et al., 2018), which is a modification of NCE (Gutmann & Hyvärinen, 2010). Following Chen et al. (2020b), InfoNCE can be defined using $2n$ instances of $n$ images in a single batch: $\mathbf{x} = [\mathcal{T}(\boldsymbol{x}_1), \mathcal{T}(\boldsymbol{x}_1), \ldots, \mathcal{T}(\boldsymbol{x}_n), \mathcal{T}(\boldsymbol{x}_n)]$, with $\mathcal{T}(\cdot)$ a stochastic image augmentation function. In this scenario, the InfoNCE loss for a single positive pair is defined as follows:

$$\mathcal{L}_{\text{InfoNCE}}(\mathbf{x}_{\{i,j\}}) = -\log \frac{\exp(\text{sim}(\boldsymbol{r}_i, \boldsymbol{r}_j))}{\sum_{k=0}^{2n} \mathbb{1}_{\{k \neq i\}} \exp(\text{sim}(\boldsymbol{r}_i, \boldsymbol{r}_k))}, \tag{1}$$

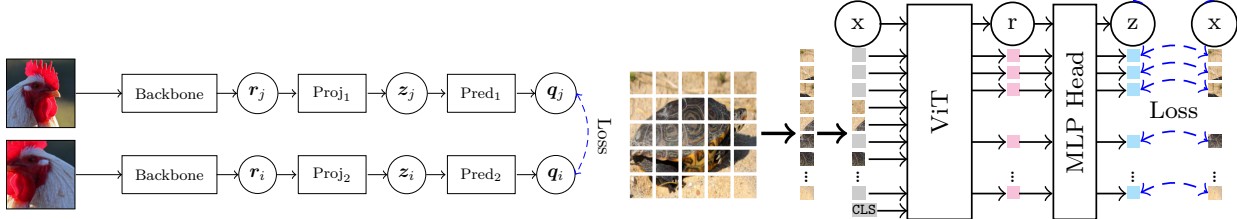

Figure 5: Illustrations of (left) dual-backbone discriminative and (right) MIM-based generative frameworks.

where $f(\mathbf{x}_i) = \boldsymbol{r}_i$ denotes the feature representation of the $i$th data point. InfoNCE is the most employed loss function for SSL frameworks that use contrastive learning (Chen et al., 2020b; He et al., 2020).

**Mean squared error** – Defined as $\mathrm{MSE}(\boldsymbol{v}, \hat{\boldsymbol{v}}) = \frac{1}{n} \sum_{i=1}^{n} (\boldsymbol{v}_i - \hat{\boldsymbol{v}}_i)^2$, the mean squared error (MSE) is employed in a number of prominent distillation-based SSL frameworks to measure feature alignment (Grill et al., 2020; Tian et al., 2021b; Caron et al., 2021). More recently, MSE has also been adopted to measure the correctness of reconstruction targets for MIM-based generative frameworks (He et al., 2022; Hou et al., 2022; Tian et al., 2023).

**Mean absolute error** – Defined as $\mathrm{MAE}(\boldsymbol{v}, \hat{\boldsymbol{v}}) = \frac{1}{n} \sum_{i=1}^{n} |\boldsymbol{v}_i - \hat{\boldsymbol{v}}_i|$, the mean absolute error (MAE) was a rarely used error measurement metric until the resurgence of MIM-based generative SSL, in which it is employed to measure the correctness of reconstruction targets (Xie et al., 2022b; Tian et al., 2023).

**Information-maximization** – Proposed by Ermolov et al. (2021); Zbontar et al. (2021); Bardes et al. (2021), a unique method of self-supervision is to maximize the information content of the embeddings (i.e., projections/predictions). Compared to the previously discussed losses, losses that maximize information content of embeddings are not only unique but also much more complicated.

For example, the loss of `VicReg` (Bardes et al., 2021) —a popular information-maximization framework— can be defined using two batches of $n$ image embeddings coming from two branches of a Siamese-like network, $\mathsf{q} = [\boldsymbol{q}_1, \ldots, \boldsymbol{q}_n]$ and $\mathsf{q}' = [\boldsymbol{q}'_1, \ldots, \boldsymbol{q}'_n]$. Then, the `VicReg` loss is defined as follows:

$$\mathcal{L}_{\mathrm{VIC}}(\mathsf{q}, \mathsf{q}') := \lambda \underbrace{s(\mathsf{q}, \mathsf{q}')}_{\text{Invariance}} + \mu \underbrace{[v(\mathsf{q}) + v(\mathsf{q}')]}_{\text{Variance}} + \nu \underbrace{[c(\mathsf{q}) + c(\mathsf{q}')]}_{\text{Covariance}}, \tag{2}$$

where $\lambda$, $\mu$, and $\nu$ are hyperparameters, and the three constituent expressions in this complex loss function play the following role: (1) The *invariance term* $s(\mathsf{q}, \mathsf{q}') = \frac{1}{n} \sum_{i=1}^{n} \|\boldsymbol{q}_i - \boldsymbol{q}'_i\|_2^2$ aims to learn invariance to data transformations by making $\mathsf{q}$ and $\mathsf{q}'$ similar. (2) The *variance term* $v(\mathsf{q})$ aims to prevent norm collapse by giving the components of $\mathsf{q}$ and $\mathsf{q}'$ a standard deviation equal to $\gamma$ (a fixed hyperparameter). It is defined as a hinge loss $v(\mathsf{q}) = \max(0, \gamma - S(\mathsf{q}, \epsilon))$, with $S(\mathsf{q}, \epsilon) = \sqrt{\mathrm{Var}(\mathsf{q}) + \epsilon}$ the regularized standard deviation. (3) The *covariance term* $c(\mathsf{q})$ strives to remove correlations between the different components of $\mathsf{q}$, and is given by the sum $\sum_{i \neq j} [C(\mathsf{q})]_{ij}^2$ over the off-diagonal elements of the $d$-dimensional covariance matrix $C(\mathsf{q})$.

## 4   Self-supervised learning frameworks

Although most recently proposed frameworks make use of a variety of techniques from both generative and discriminative SSL, the frameworks that we will discuss shortly can be typically categorized as either generative or discriminative. In the case when a framework leverages techniques that belong to multiple categories and may thus fall into more than one category, we adopt the designation used by its creators. Since most of the frameworks are known by their acronyms, we use their abbreviated names in the main text and provide their full names in Section A of the the appendix.

| SSL framework | Proposed by | Unique property |
|---|---|---|
| `Deep Cluster` | Caron et al. (2018) | Avoids trivial solutions for clustering-based SSL |
| `Local Aggregation` | Zhuang et al. (2019) | Local aggregation metric for soft cluster assignments |
| `Deeper Cluster` | Caron et al. (2019) | Integrates rotation-based SSL into clustering |
| `SeLa` | Asano et al. (2019) | Improves `Deep Cluster` with the Sinkhorn-Knopp algorithm |
| `SCAN` | Van Gansbeke et al. (2020) | Decouples feature learning and clustering using a two-step approach |
| `Deep Cluster-v2` | Caron et al. (2020) | Incorporates various SSL improvements into `Deep Cluster` |
| `SeLa-v2` | Caron et al. (2020) | Incorporates various SSL improvements into `SeLa` |
| `Swav` | Caron et al. (2020) | Online clustering with consistency across assignments |
| `ODC` | Zhan et al. (2020) | Converts `Deep Cluster` into an online method |
| `CoKe` | Qian et al. (2022) | Improves the clustering phase with an online constrained k-means method |
| `Self-Classifier` | Amrani & Bronstein (2021) | Single-stage end-to-end clustering combined with contrastive learning |

Table 1: SSL frameworks that rely on **clustering**-based self-supervision and their unique properties.

## 4.1 Discriminative SSL

In terms of discriminative SSL, frameworks can roughly be grouped by their reliance on the following techniques: **clustering**, **contrastive learning**, **distillation** and **information-maximization**. In what follows, we detail discriminative SSL frameworks that fall under the aforementioned categories.

### 4.1.1 Clustering

Self-labeling via clustering is one of the most straightforward ways to achieve self-supervision, with clustering being one of the most popular methods for unsupervised learning (Bishop, 2006). For neural networks, the usage of clustering-based methods for training can be traced back to the seminal works of Coates et al. (2011), Coates & Ng (2012), and Yang et al. (2016), which paved the way for the use of such methods for SSL. Unfortunately, clustering-based methods have to solve a number of well-documented issues such as: (1) offline training that prevents their usage for large-scale data, (2) large clusters dominating the majority of the labels or small clusters leading to extremely granular labels, (3) empty clusters, (4) requiring knowledge about the number of clusters beforehand, and (5) trivial solutions where all data are gathered in a single cluster which causes the network to collapse (Xu et al., 2004; Joulin et al., 2016). Since these issues are fundamental problems of clustering, all of the clustering-based SSL methods have to tackle these problems in their own unique way when trying to perform self-supervision.

The pioneering work of Caron et al. (2018) put forward `Deep Cluster`, one of the first clustering-based SSL methods that achieves results comparable to supervised models. This method solves the issues listed above with an offline training approach and by forcing a uniform distribution across clusters, both of which limit the usage of `Deep Cluster`. Following that, getting rid of the tricks applied in `Deep Cluster` became the primary focus of a number of subsequent studies, leading to improved clustering-based SSL methods such as `SeLA` (Asano et al., 2019), `Online Deep Cluster` (Zhan et al., 2020), and `Self-Classifier` (Amrani & Bronstein, 2021). `SeLa` tackles the issue of model collapse by incorporating a more principled loss using the Sinkhorn-Knopp algorithm (Cuturi, 2013). `Online Deep Clustering` on the other hand addresses the aforementioned offline training issue to enable online training for large datasets.

Conversely, Van Gansbeke et al. (2020) argue that an end-to-end approach with online training may lead to various problems and propose an approach called `SCAN` that replaces the use of K-Means for the purpose of clustering with the use of an advanced neighbor search. When it comes to the state-of-the-art, the clustering-based method proposed in Caron et al. (2020), known as `Swav`, which also leverages a number of contrastive elements, is currently considered to be the most stable and accurate approach. Table 1 provides a summarizing overview of several clustering-based SSL methods, detailing their unique traits.

### 4.1.2 Contrastive learning

Contrastive learning with the InfoNCE loss (or an extension of it) is the most popular approach for self-supervision and also the one that received the most research contributions in the past years. Contrastive methods can be traced back to the works of Bromley et al. (1993) and Chopra et al. (2005), but in terms of modern usage of SSL, Wu et al. (2018b) and Oord et al. (2018) popularized this line of research by proposing `InstDist` and `CPC`, respectively. Hjelm et al. (2018) and Bachman et al. (2019) investigated different ways to

| SSL framework | Proposed by | Unique property |
|---|---|---|
| InstDist (NPID) | Wu et al. (2018b) | Non-parametric softmax calculation |
| CPC | Oord et al. (2018) | Usage of InfoNCE loss across multiple tasks |
| DIM | Hjelm et al. (2018) | Measures representation quality with two novel losses (MINE and NDM) |
| CPC-v2 | Henaff (2020) | Improves CPC architecture and training |
| AMDIM | Bachman et al. (2019) | Extends DIM for mixture-based representations |
| CMC | Tian et al. (2020a) | Information-maximization across different sensory views |
| MoCo | He et al. (2020) | SSL with momentum encoder and memory bank |
| PIRL | Misra & Maaten (2020) | Contrastive learning with jigsaw puzzles |
| SimCLR | Chen et al. (2020b) | Usage of projection heads and new augmentations |
| MoCo-v2 | Chen et al. (2020d) | Improves MoCo with the design of SimCLR |
| SimCLR-v2 | Chen et al. (2020c) | Improves SimCLR with memory bank and deeper projector MLPs |
| PCL & PCL-v2 | Li et al. (2020b) | Formulates contrastive learning with clustering using EM |
| PIC | Cao et al. (2020) | One-branch parametric instance classification |
| DCL | Chuang et al. (2020) | Negative sample section with a debiased contrastive objective |
| LooC | Xiao et al. (2020) | Learns transformation dependent and invariant representations |
| G-SimCLR | Chakraborty et al. (2020) | SimCLR with negative sample selection using pseudo-labels |
| ReLIC | Mitrovic et al. (2020) | Imposes invariance constraints during SSL training |
| AdCo | Hu et al. (2021) | Mixes self-trained negative adversaries into SSL |
| DenseCL | Wang et al. (2021c) | Dense contrastive loss for SSL |
| PixPro | Xie et al. (2021c) | PixContrast and PixPro losses for contrastive SSL |
| MoCo-v3 | Chen et al. (2021) | Improves MoCo-v2 with symmetrized loss and without a memory bank |
| CLSA | Wang & Qi (2022) | Usage of stronger augmentations for contrastive learning |
| Truncated Triplet | Wang et al. (2021b) | Attempts to solve under- and over-clustering in contrastive learning |
| NNCLR | Dwibedi et al. (2021) | Nearest-neighbors as positive samples in contrastive loss |
| MoBY | Xie et al. (2021b) | Combines design principles of MoCo and BYOL for transformers |
| DNC | Tian et al. (2021a) | Alternation of contrastive learning and clustering-based hard negative mining |
| ReSSL | Zheng et al. (2021) | Maintains the relational consistency between different instances of images |
| UniGrad | Tao et al. (2022a) | Unifies contrastive learning, distillation, and information-maximization |
| ReLIC-v2 | Tomasev et al. (2022) | Improves ReLIC with inductive biases to learn more informative representations |
| SimCo | Zhang et al. (2022a) | Simplifies MoCo with momentum removal |
| SimMoCo | Zhang et al. (2022a) | Simplifies MoCo with dictionary removal |
| UniVIP | Li et al. (2022b) | Scene-based SSL based on similarity, correlation, and discrimination |
| Mugs | Zhou et al. (2022) | Explicitly learns multi-granular visual features |
| CaCo | Wang et al. (2022b) | Learns both positive and negative samples end-to-end with an encoder |
| SMoG | Pang et al. (2022) | Replaces instance contrastive learning with group contrastive learning |
| SiameseIM | Tao et al. (2022b) | Instance discrimination using UniGrad and masked images |

Table 2: SSL frameworks that rely on **contrastive learning**-based self-supervision and their unique properties.

measure representation quality for contrastive learning and proposed DIM and AMDIM respectively, while Tian et al. (2020a) extended contrastive learning for multiple sensory inputs with CMC. After the aforementioned works contrastive SSL attracted significant research interest but it was the groundbreaking results obtained with MoCo which used memory banks with delayed weight updates that put contrastive SSL really into the spotlight (He et al., 2020). Shortly after, Chen et al. (2020b) proposed SimCLR and with it, further improved the state-of-the-art with the help of projection heads and strong augmentations and cemented the importance of contrastive self-supervision as a learning paradigm. Incorporating the enhancements of SimCLR into MoCo, Chen et al. (2020d) proposed MoCo-v2 and showed that there still exists a large margin for improvement. Chen et al. (2021) later introduced a third version of MoCo, exploring the usage of vision transformers as backbones. The reliable design of MoCo and its improved versions were the foundation of many subsequent contrastive SSL frameworks, such as AdCo (Hu et al., 2021), MocHi (Kalantidis et al., 2020), and DenseCL (Wang et al., 2021c).

While the above architectures mostly use dual backbones, Cao et al. (2020) proposed PIC and demonstrated the viability a single-branch backbone architecture for contrastive learning. Kalantidis et al. (2020) experimented with hard negative samples for improving the effectiveness of contrastive learning and Wang & Qi (2022) demonstrated the usefulness of stronger augmentations. After the success of Moco-v2 and Moco-v3, and with the increased availability of unique SSL methods, frameworks like G-SimCLR (Chakraborty et al., 2020), MoBY (Xie et al., 2021b), SimCo, and SimMoCo (Zhang et al., 2022a), which combine multiple SSL methods into a single one, gained traction. More recently, SSL frameworks such as UniGrad (Tao et al., 2022a) claim to combine four self-supervision methodologies (clustering, contrastive, distillation, and information-maximization) into a single framework and to unify discriminative SSL training.

Although contrastive methods garnered more attention than clustering-based methods, they are also subject to a similar problem that needs to be mitigated: network collapse (Jing et al., 2021). Contrastive

| SSL framework | Proposed by | Unique property |
|---|---|---|
| BYOL | Grill et al. (2020) | Avoids trivial solutions through network asymmetry |
| SimSiam | Chen & He (2021) | SSL with simple Siamese networks without negative samples |
| OBoW | Gidaris et al. (2021) | Online bag-of-visual-words for SSL |
| DirectPred | Tian et al. (2021b) | Adjusts linear predictor with a gradient-free approach |
| SEED | Fang et al. (2021) | Knowledge distillation from large to small models |
| DisCo | Gao et al. (2021) | Combines contrastive and distillation learning for lightweight models |
| DINO | Caron et al. (2021) | Knowledge distillation with vision transformers |
| EsViT | Li et al. (2021a) | Multi-stage architectures with sparse self-attentions and region matching for efficient SSL |
| BINGO | Xu et al. (2021) | Distillation-based SSL for small-scale models |
| TinyMIM | Ren et al. (2023) | Distillation to transfer knowledge from large MIM-based models to small models |

Table 3: SSL frameworks that rely on **distillation**-based self-supervision and their unique properties.

| SSL framework | Proposed by | Unique property |
|---|---|---|
| WMSE | Ermolov et al. (2021) | Whitening Mean Squared Error loss for information-maximization |
| Barlow Twins | Zbontar et al. (2021) | SSL with redundancy reduction |
| VicReg | Bardes et al. (2021) | Variance-invariance-covariance regularization for avoiding collapse |
| TWIST | Wang et al. (2021a) | Theoretically explainable TWIST loss that avoids collapse |
| TLDR | Kalantidis et al. (2021) | Improves Barlow Twins with TLDR encoder |
| ARB | Zhang et al. (2022b) | Aligns feature representations with nearest orthonormal basis |
| VicRegL | Bardes et al. (2022) | Improves VicReg with location- and feature-based matching |

Table 4: SSL frameworks that rely on **information-maximization**-based self-supervision and their unique properties.

methods prevent complete collapse of a network through the use of negative samples. However, Hua et al. (2021) surprisingly demonstrated that contrastive SSL frameworks can suffer from another type of collapse, namely dimensional collapse, wherein representations collapse into a low-dimensional manifold. Given the importance of negative samples in preventing collapse in contrastive SSL, understanding the effects of negative samples and finding better sampling techniques became an active research topic shortly after (Chuang et al., 2020; Robinson et al., 2020; Zhang et al., 2022a). A summarizing overview of several contrastive SSL frameworks can be found in Table 2.

### 4.1.3 Distillation

Can the collapse of networks be prevented without the use of self-labeling or a contrastive loss that relies on negative samples? Through an asymmetric framework called BYOL, Grill et al. (2020) demonstrated that neither of those techniques are necessary to achieve self-supervision when the proposed method relies on distillation (Hinton et al., 2015). The general idea behind distillation is to train a network (student) to predict representations of another one (teacher) (Tarvainen & Valpola, 2017). Shortly after the proposal of BYOL, Chen & He (2021) proposed SimSiam, a symmetric (Siamese) framework that uses neither negative samples nor clustering, but leverages instead stop-grad and projection/prediction MLPs. This was followed by OBOW (Gidaris et al., 2021), in which the task is to reconstruct a bag-of-visual-words representation.

Similar to the trends witnessed for clustering and contrastive-learning, distillation-based SSL frameworks were experimentally combined with other frameworks in an attempt to obtain boosts in overall effectiveness. Frameworks such as DisCo (Gao et al., 2021) and MoBY (Xie et al., 2021b) merged multiple frameworks together, while others tried to improve the effectiveness of established methods, such as MSF (Koohpayegani et al., 2021) and ORL (Xie et al., 2021a), improving upon BYOL.

How do distillation methods avoid network collapse? Tian et al. (2020c) and Fetterman & Albrecht (2020) argued that methods that incorporate batch statistics into training (e.g., batch normalization) aid BYOL (and potentially other distillation-based methods) in preventing collapse, but this hypothesis was promptly refuted by Richemond et al. (2020). Recently, Li et al. (2022a) scrutinized SimSiam and found it to be highly sensitive to model size. Nevertheless, a definite answer to the way distillation-based SSL methods avoid collapse is not yet found. Table 3 provides a summarizing overview of several SSL frameworks that rely on distillation.

| Module | Proposed by | Unique property |
|---|---|---|
| InfoMin | Tian et al. (2020b) | InfoMin principle and evaluation of augmentations |
| InterCLR | Xie et al. (2022a) | Inter-image invariance for contrastive learning |
| HEXA | Li et al. (2020a) | Proposes new data augmentation methods that are harder to predict |
| MocHi | Kalantidis et al. (2020) | Hard negative image mixing approach |
| ReSim | Xiao et al. (2021) | Enhances SSL representations with region similarities |
| MSF | Koohpayegani et al. (2021) | Enhances BYOL by shifting the embeddings to be close to the mean of its instances |
| ORL | Xie et al. (2021a) | Utilizes BYOL for object-level training |
| CEB | Lee et al. (2021) | Measures the amount of compression in the learned representations |
| SEM | Lavoie et al. (2022) | Employs simplicial embeddings to map unnormalized representations onto simplices |
| ENS | Ruan et al. (2022) | Investigates optimal ensemble models for discriminative SSL frameworks |
| MRCL | Liu et al. (2022b) | Uses MIM as a method to avoid the discriminative information overfitting |
| TS | Kukleva et al. (2023) | Assists contrastive methods to learn group-wise features and instance-specific details |
| ARCL | Zhao et al. (2023) | Enhances contrastive learning with domain-invariant features representations |
| MosRep | Wang et al. (2023) | Data augmentation strategy that enriches backgrounds of crops |

Table 5: **Enhancements** for existing discriminative SSL frameworks and their unique properties.

### 4.1.4 Information-maximization

The fourth and final discriminative self-supervision category we cover is information-maximization, having as primary idea the maximization of the information conveyed by decorrelated embeddings. Such approaches come with a number of advantages, in particular, they neither require negative samples nor require an asymmetric architecture to avoid collapse. Instead, they completely rely on innovative loss functions to avoid collapse. As a result, most of the frameworks that fall under this category can be characterized by the novel loss function that is used.

Information-maximization as a method for self-supervision was put forward by Ermolov et al. (2021) and Zbontar et al. (2021), where the former proposed W-MSE loss, which constrains the batch samples to dissipate in a spherical distribution, and where the latter (Barlow Twins) aims at making the normalized cross-correlation matrix of the embedding vectors to be close to the identity matrix. Bardes et al. (2021) further improved the loss of Barlow Twins with the VicReg framework, proposing a loss based on variance, invariance, and covariance (described above in equation 2). Successor frameworks such as TWIST (Wang et al., 2021a), TLDR (Kalantidis et al., 2021), and ARB (Zhang et al., 2022b) followed the path paved by the previous frameworks and aim at improving the losses in different ways. Due to the complex nature of the losses used in information-maximization as a method for self-supervision, we refer the interested reader to the respective research papers underlying those frameworks. Table 4 provides a summarizing overview of several SSL frameworks that rely on information-maximization.

### 4.2 Enhancements for existing frameworks

So far, we have covered a large number of discriminative SSL frameworks, all of which have consistently improved state-of-the-art results across various computer vision tasks. However, we have observed a trend that emerged towards the end of 2020: framework-agnostic enhancements. These modules are small tweaks to existing frameworks that can improve their performance in various ways, such as utilizing harder images/augmentations (Kalantidis et al., 2020; Li et al., 2020a), improving object-level representations (Xiao et al., 2021; Xie et al., 2021a), or enabling optimal ensemble models (Ruan et al., 2022). For completeness, we have listed these modules separately in Table 5.

### 4.3 Generative SSL

From 2018 onward, generative SSL was largely dismissed in favor of discriminative training methods with contrastive learning holding the prime spot for research (He et al., 2020; Chen et al., 2020b). Popular discriminative frameworks such as MoCo, SimCLR, and BYOL were employed for a variety of unique tasks and were further improved with enhancements taken from each other (Chen et al., 2020d;c). In an unexpected turn of events, towards the end of 2021, generative SSL came to dominate image-based self-supervised learning and became the primary research focus, dethroning contrastive learning as well as discriminative SSL (Bao et al., 2021; Zhou et al., 2021). The advances in the field from the last quarter of 2021 to the first quarter of 2023 were so rapid that state-of-the-art results in generative SSL were improved on a

| SSL framework | Proposed by | Unique property |
|---|---|---|
| `BiGAN` | Donahue et al. (2016) | Bidirectional `GAN` with additional encoder modela |
| `ALI` | Dumoulin et al. (2016) | A `GAN` framework with an additional inference model |
| `BigBiGAN` | Donahue & Simonyan (2019) | `BiGAN` with the generator of `BigGAN` |
| `SS-GAN` | Chen et al. (2019) | `GAN` with auxiliary rotation loss |
| `SS-GAN-LA` | Hou et al. (2021) | `SS-GAN` with label augmentation |
| `Vit-VQGAN` | Yu et al. (2021) | `VQGAN` with label quantization and ViT backbone |

Table 6: SSL frameworks that rely on **GAN-based** generative self-supervision and their unique properties.

monthly basis. This rapid expansion also came with a large category of unique approaches which resulted in frameworks of generative SSL becoming much less standardized as opposed to frameworks in discriminative SSL where the latter mostly contains straightforward Siamese-like dual-backbones as shown in Figure 5. In order to improve readability, we will group generative SSL frameworks into two categories: the ones that use generative adversarial networks (`GAN`s) and others that use a form of masked image modeling.

### 4.3.1 GAN-based generative SSL

While the usage of generative neural networks can be traced back to the work of Hinton et al. (2006), it was the seminal work of Goodfellow et al. (2020) that popularized generative models with the newly proposed `GAN` framework. Since the work of Goodfellow et al. (2020), numerous `GAN` variants were proposed with some of them recently taking advantage of advances in SSL, such as incorporating rotation prediction (Chen et al., 2019), jigsaw puzzles (Baykal & Unal, 2020; Baykal et al., 2022), and self-labeling (Lučić et al., 2019). However, most of the research in the `GAN` space has primarily focused on enhancing the fidelity of images generated by the generator network, which is typically evaluated using metrics such as the Fréchet Inception Distance (Heusel et al., 2017). As a result, these studies largely ignore the discriminative network and lack comparative evaluations on downstream tasks, leaving them out of the scope of SSL. In what follows, we focus on those research efforts that evaluate the discriminative power of `GAN`s on downstream tasks.

With a unique twist to `GAN`s, Donahue et al. (2016) proposed `BiGAN`, a `GAN` framework that contains an additional encoder network trained in conjunction with the generator and discriminator networks with the objective of inverting the generator. After the training is completed, this encoder can be used as a feature extractor for downstream tasks. Independently, Dumoulin et al. (2016) proposed a generative framework called `ALI` that is almost identical to `BiGAN`. Leveraging the improved generator of `BigGAN` (Brock et al., 2018) in `BiGAN`, Donahue & Simonyan (2019) proposed `BigBiGAN` which comes with better downstream transferability results. Taking inspiration from the developments in the area of discriminative SSL, Chen et al. (2019) proposed `SS-GAN` which exploits rotation as an auxiliary task to achieve self-supervision with `GAN`s. This framework was further improved with the addition of label augmentation by Hou et al. (2021). One of the most recent approaches within `GAN`-based SSL is `ViT-VQGAN` (Yu et al., 2021) which improves `VQGAN` (Esser et al., 2021) using `ViT` backbones.

Overall, the usage of `GAN`s in SSL has not become a mainstream method due to a number of `GAN`-related limitations, ranging from mode collapse to limitations related to scalability, as well as lack of flexibility in backbone networks.

### 4.3.2 Generative SSL with masked image modeling

Although only a couple of years have passed since the discovery of ViTs (Dosovitskiy et al., 2020), these architectures have been shown to achieve state-of-the-art results in a variety of vision tasks. In their work, Dosovitskiy et al. (2020) demonstrated the feasibility of SSL using MIM as a pretext task (although it was called masked patch prediction by the authors), where this pretext task is seamlessly supported by the patch-based image intake of ViTs. Developing this technique further, `BEiT` was one of the first frameworks that successfully employed MIM with vector quantized images and ViTs (Bao et al., 2021). It is important to note that `BEiT` does not directly predict the pixel values of the image but learns to predict discrete visual tokens which are created from image patches (Wu et al., 2020). `BEiT` uses the tokenizer — a discrete variational autoencoder (dVAE) — of `DALL-E` (Ramesh et al., 2021) which requires an offline training before training the final

| SSL framework | Proposed by | Unique property |
|---|---|---|
| iGPT | Chen et al. (2020a) | MIM with 9-bit pixel clustering per patch |
| BEiT | Bao et al. (2021) | Patch-based MIM with ViTs using offline DALL-E tokenizer |
| MAE | He et al. (2022) | MIM with autoencoders using lightweight ViT encoders and pixel-based reconstruction |
| iBOT | Zhou et al. (2021) | BEiT with an online tokenizer trained using the DINO objective |
| SimMIM | Xie et al. (2022b) | BEiT with a pixel reconstruction target without a tokenizer |
| PeCO | Dong et al. (2021) | Proposes a new codebook to replace the DALL-E tokenizer |
| MaskFeat | Wei et al. (2022) | MIM training with HOG as a reconstruction target |
| data2vec | Baevski et al. (2022) | BEiT with a teacher-student model and stronger augmentations |
| CAE | Chen et al. (2022a) | MAE with DALL-E token target reconstruction |
| CIM | Fang et al. (2023) | Corrupted image modeling for generative SSL with an additional discriminative objective |
| MCMAE | Gao et al. (2022) | Multi-scale hybrid convolution-transformer for improved MIM performance |
| ConMIM | Yi et al. (2022) | Contrastive learning on MIM patches |
| CMAE | Huang et al. (2022) | MIM + contrastive learning with shifted image views |
| SdAE | Chen et al. (2022b) | Self-distillation with high-level feature reconstruction |
| MILAN | Hou et al. (2022) | MIM with semantic-aware mask sampling and CLIP-assisted feature reconstruction |
| BEiT-v2 | Peng et al. (2022) | BEiT with CLIP tokenizer as a teacher and patch aggregation strategy |
| BEiT-v3 | Wang et al. (2022a) | BEiT-v2 with text fusion |
| CAE-v2 | Zhang et al. (2022d) | CAE with CLIP tokenizer |
| CAN | Mishra et al. (2023) | Combines contrastive learning, MIM, and image denoising with symmetric backbones |
| PCAE | Li et al. (2023) | Progressively drops reconstruction tokens in MAE for better speed/performance trade-off |
| SparK | Tian et al. (2023) | MIM for convolutional neural networks |
| MRMAE | Gao et al. (2023) | Uses pixels, DINO features, and CLIP features for reconstruction |

Table 7: SSL frameworks that rely on **MIM-based** generative self-supervision and their unique properties.

model. BEiT gave rise to BEiT-v2 (Peng et al., 2022) and BEiT-v3 (Wang et al., 2022a)[1] which obtain better results using the CLIP tokenizer (Radford et al., 2021) with a patch aggregation strategy. Meanwhile, the requirement of an external tokenizer for BEiT was alleviated by iBOT (Zhou et al., 2021) which introduced an online tokenizer trained with the distillation routine of BYOL, thus leveraging the advances made on the side of discriminative SSL. Xie et al. (2022b) got rid of the tokenizer of BEiT and proposed SimMIM, which directly operates over pixel values and predicts them.

Narrowly predating BEiT, Chen et al. (2020a) proposed iGPT by leveraging GPT-2 (Radford et al., 2019) and adapted it to vision, which represents images with tokenized patches using a 9-bit color palette by clustering RGB pixels, and then training this model with the MLM objective of BERT (Devlin et al., 2018). The primary difference between the training objective of iGPT (i.e., BERT-style MLM) and MIM of BEiT is that the latter directly uses image patches as an input, therefore not losing any pixel-level information.

With a unique take on MIM, He et al. (2022) proposed MAE, an asymmetric autoencoder framework that directly learns to reconstruct image patches. What is unique to MAE is that its encoder (ViT) only processes unmasked patches (e.g., 25% of all patches) without any tokenizer, making it much faster than the frameworks we have discussed thus far. He et al. (2022) also evaluated the effectiveness of different reconstruction targets and found no statistically significant difference between reconstructing DALL-E tokens and pixels, suggesting simple pixel reconstruction to be a viable reconstruction target. Building upon MAE, Chen et al. (2022a) proposed CAE which comes with a latent contextual regressor and uses the DALL-E tokenizer, which was replaced in the next iteration of this framework (CAE-v2) (Zhang et al., 2022d) by a CLIP tokenizer.

At this point we believe it is important to reiterate that MIM was explored using different reconstruction targets such as (1) dVAE-based patch tokens (Bao et al., 2021), (2) clustering-based patch tokens (Chen et al., 2020a), and (3) pixel values (Xie et al., 2022b). Expanding this corpus, Wei et al. (2022) proposed MaskFeat in which the task is to predict Histograms of Oriented Gradients (HOGs) — a hand-crafted feature descriptor (see Figure 2i) — and argued that a broad spectrum of image features can be used as targets in MIM. Following their work, SdAE (Chen et al., 2022b) and MILAN (Hou et al., 2022) demonstrated the feasibility of reconstructing high-level features. For an overview of reconstruction targets for MIM-based generative SSL frameworks, see Table 8.

---

[1] BEiT-v3 is a framework that fuses vision and text but we include it for completeness.

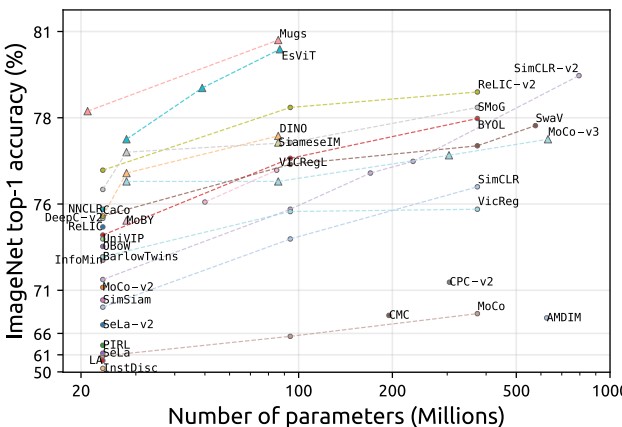

Figure 6: ImageNet top-1 accuracy with **linear probing** on frozen representations for **discriminative SSL frameworks** is plotted against the number of parameters in the trained backbone. The connecting lines indicate different backbone networks trained with the same framework. In both figures, nodes with circles indicate CNN-based architectures, whereas triangles indicate transformer-based architectures. Note that a few frameworks with overlapping results are omitted from the figures, and that axis values are scaled independently to improve visual clarity.

Table 8: Reconstruction targets for generative SSL frameworks that use MIM.

| Framework | Reconstruction |
|-----------|----------------|
| iGPT | 9-bit pixels |
| BEiT | DALL-E tokens |
| MAE | Pixels |
| iBOT | Distilled tokens |
| SimMIM | Pixels |
| PeCO | PeCO tokens |
| MaskFeat | HOG |
| data2vec | DALL-E tokens |
| CAE | DALL-E tokens |
| CIM | DALL-E tokens |
| MCMAE | Pixels |
| ConMIM | Patch features |
| CMAE | Pixels |
| SdAE | High-level features |
| MILAN | High-level features |
| BEiT-v2 | CLIP tokens |
| BEiT-v3 | CLIP tokens |
| CAE-v2 | CLIP tokens |
| CAN | Pixels |
| PCAE | Pixels |
| SparK | Pixels |
| MRMAE | CLIP tokens |

Experimenting with the input side Fang et al. (2023) proposed `CIM` where an auxiliary generator (in their use-case, `BEiT`) corrupts the input image and the proposed framework aims to (1) discriminate patches (classification for each patch) and (2) generate the original image.

The current trend for generative SSL is to combine both generative and discriminative losses together to improve the quality of representations. In particular, contrastive learning has become a popular task to combine with MIM with frameworks such as `CAN` (Mishra et al., 2023), `CMAE` (Huang et al., 2022) and `ConMIM` (Yi et al., 2022) leveraging advances made on the side of contrastive SSL.

All of the generative frameworks we have discussed thus far use some form of vision transformer as a backbone as opposed to the majority of the discriminative frameworks, which make use of ResNets. In an attempt to leverage masked convolutions, Gao et al. (2022) proposed `MCMAE`, a framework that employs the hybrid convolution-transformer `MAE` which is able to learn discriminative representations. Very recently, Tian et al. (2023) showed that classical (ResNets) and modern (ConvNext (Liu et al., 2022a)) CNNs can be trained with MIM and achieve state-of-the-art results that can rival those of ViTs. The results obtained by Tian et al. (2023) indicate that ViTs, which have been considered a prerequisite for MIM, are not irreplaceable and that CNNs can still compete with ViTs in generative SSL.

## 5 Evaluating SSL models

As briefly noted in Section 3, the SSL frameworks covered thus far are concerned with the training of feature extractors that can extract robust and useful features from images. Regardless, those feature extractors must be evaluated for a fair comparison of performance, which is the focus of this section.

In the SSL literature, three types of evaluations are commonly used: (i) fine-tuning the entire model, (ii) linear evaluation, also known as linear probing or linear protocol, and (iii) K-nearest neighbors (KNN) evaluation using extracted features. A further distinction can be made based on the dataset the trained model is evaluated on: either (a) on the same dataset, typically ImageNet, that was used for the self-supervised training or (b) on different datasets to test downstream transferability.

**KNN evaluation** – After the SSL training is complete, features of the training images are generated from the backbone and matched to their corresponding labels, thus creating a feature bank. Next, predictions are made for the test images based on the KNN labels of this feature bank. For KNN-based evaluation, following Wu et al. (2018b), $k = 200$ is often used. Although this form of evaluation was popular early on, the majority of recently proposed frameworks evaluate their models using linear evaluation or by fine-tuning.

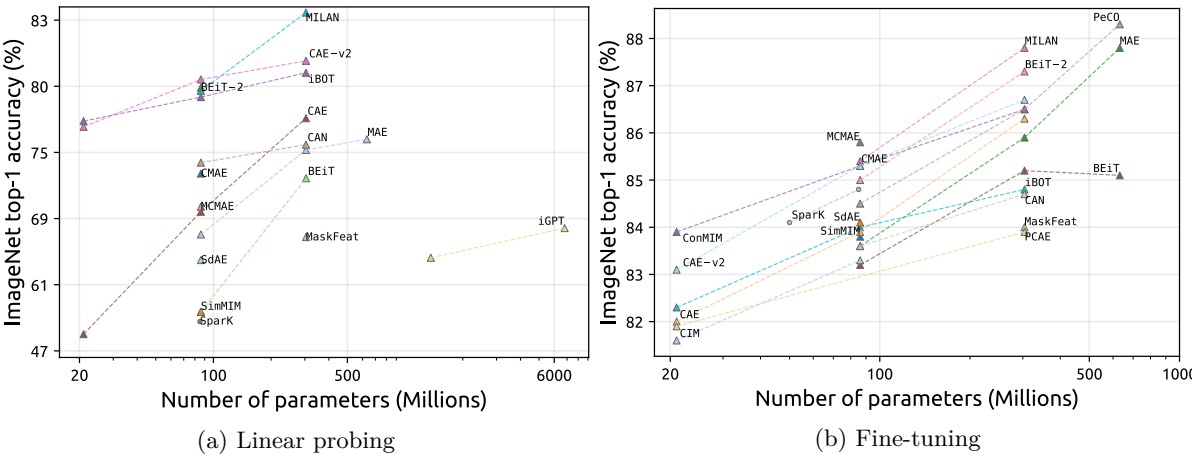

(a) Linear probing
(b) Fine-tuning

Figure 7: The ImageNet Top-1 accuracy of backbones that are trained with MIM-based generative SSL frameworks are measured using (a) **linear probing** and (b) **fine-tuning**. The connecting lines indicate different backbone networks trained with the same framework. In both figures, nodes with circles indicate CNN-based architectures, whereas triangles indicate transformer-based architectures.

**Linear evaluation** – For this type of evaluation, all trainable parameters (e.g., weights) in the model are frozen and a new linear layer, which maps features to predictions, is introduced to the trained model. Then, only this linear layer is trained on the training set to achieve an optimal performance.

**Fine-tuning** – For this type of evaluation, once again, a linear layer is introduced to the SSL-trained model which maps features to predictions. Then, in a supervised fashion, the entire model is (re)trained on the training dataset, after which an evaluation is performed on the test/validation set.

**Benchmarks** – In order to provide an aggregate view of the field, we provide the benchmarking results below.

- For the majority of the discriminative SSL frameworks covered in this survey, we provide a comparison of model size to linear probing accuracy on ImageNet in Figure 6.

- For MIM-based generative SSL frameworks, we provide a comparison of model size to linear probing and fine-tuning accuracy on ImageNet in Figure 7.

- From Table 21 to Table 23, we provide the datasets used in the respective papers of SSL frameworks.

- From Table 24 to Table 30, we provide benchmarks on ImageNet-1K.

- In Table 31, we provide benchmarks on COCO.

**Evaluation preference for discriminative and generative frameworks** – A noteworthy observation in the evaluation of self-supervised learning frameworks is that while most discriminative SSL frameworks tend to favor linear evaluation, the majority of generative SSL frameworks tend to prefer fine-tuning. This is primarily due to the poor results obtained with linear probing with generative frameworks that only use MIM as the pretext task (see `MAE`, `BEiT`, `SimMIM`, `iGPT` in Figure 7). When discriminative elements such as contrastive learning, distillation, or the `CLIP` tokenizer which is trained contrastively, are used, linear probing accuracy shows a dramatic increase (see `MILAN`, `iBOT`, `CAE-v2`, and `BEiT-v2` in Figure 7).

## 6  Availability and comparability of SSL frameworks

Most of the frameworks covered in Section 4 perform experiments on ImageNet, COCO (Lin et al., 2014), and Pascal VOC (Everingham et al., 2007), thus enabling straightforward benchmarking and comparability. Moreover, many SSL frameworks come with implementations and trained models that are publicly available,

contributing to speeding up research on SSL. For example, the availability and the straightforward adoptability of the `MoCo` framework enabled a number of follow-up studies that used the code of `MoCo` (Kalantidis et al., 2020; Hu et al., 2021; Wang et al., 2021c). For the SSL frameworks covered in this survey, in Table 17, Table 18, and Table 19 we provide details on the availability of official implementation as well as trained models.

Apart from the availability of official implementations, the availability of third-party repositories also accelerated the adoption of SSL, enabling unified experimentation. Alas, not all third-party repositories are up to date, and some of them have already been abandoned. In Table 20, we provide a number of useful SSL repositories that have been updated within the third quarter of 2022.

## 7 Conclusions, current trends, and directions for future research

In this survey, we reviewed general-purpose frameworks that use images for SSL training, with the goal of bringing interested researchers up to speed with the field of SSL. In what follows, we highlight a number of directions for future research that are ripe for contribution.

**Theoretical understanding of the requirements of SSL** – As detailed in Section 4, the successful implementation of discriminative self-supervised learning frameworks requires several prerequisites. To this end, several studies have investigated the efficacy of these requirements, covering topics such as the necessity of negative samples (Kalantidis et al., 2020; Xie et al., 2022a), the importance of image augmentations (Zhang et al., 2022c; Wang et al., 2022d), and architectural tricks (Chen & He, 2021). Furthermore, a number of research efforts have attempted to explain the underlying mechanisms behind collapse avoidance (Garrido et al., 2022b; Chen et al., 2023b). Because they were the earliest self-supervision methods, clustering (Assran et al., 2023) and contrastive learning (Hu et al., 2023; Johnson et al., 2023; Tian, 2023) have received significant attention in terms of theoretical contributions. However, other self-supervision paradigms are areas where theoretical explanations are still lacking and are open for further research efforts.

**Domain- and task-specific SSL** – The majority of the frameworks covered in this survey are task-agnostic and evaluate their performance on the ImageNet dataset and a number of various downstream tasks focusing on natural images. However, the effectiveness of these models on natural image datasets may not necessarily generalize to other datasets that contain different image modalities or to other tasks. Therefore, investigating the effectiveness of SSL frameworks that leverage the unique characteristics of data in other domains such as medical imaging (Ramesh et al., 2022; Chen et al., 2023a) or other tasks such as classification in the wild (Goyal et al., 2021a; Tian et al., 2021a), object detection (Mishra et al., 2021; Li et al., 2022b), pose estimation (Chen et al., 2023c), action detection as well as human-object interaction (Wei et al., 2022; Shah et al., 2023) represents a relevant area of research.

**Calibration, interpretability, and adversarial robustness** – Initial findings suggest that models trained using SSL exhibit distinct properties for robustness and interpretability when compared to models trained via supervised learning (Hendrycks et al., 2019; Zhong et al., 2022). However, many of the beneficial and detrimental effects of self-supervised training on downstream tasks remain unclear.

**Efficient SSL** – The training of SSL models demands a substantial amount of computational resources in comparison to supervised learning. For instance, as reported by Chen et al. (2021), the training of `MoCo-v3` with a vision transformer backbone requires approximately 625 TPU days. Consequently, SSL has significantly increased the computational demands of DNN training. This observation explains why a vast majority of the contributions to the frameworks discussed in this survey have at least one author with an industry affiliation (see Table 17 to Table 19). Moreover, the majority of these contributions ($> 80\%$) come from industry labs such as Facebook AI Research, Microsoft Research, DeepMind, Google Research, SenseTime, ByteDance, and Huawei. To mitigate the high costs of training, researchers have started exploring techniques for efficient training and evaluation (Li et al., 2021a; Garrido et al., 2022a; He et al., 2022). Despite the progress made, there is still a considerable amount of work to be done in this field.

**KNN-based evaluation, linear probing, or fine-tuning?** – As we mentioned in Section 5, most generative SSL frameworks prefer to use fine-tuning as the method of evaluation while discriminative frameworks prefer to use linear probing. In favor of those two, KNN-based evaluation has been mostly abandoned.

Chen & He (2021) and He et al. (2022) argue that there is no correlation between the accuracy of linear probing and fine-tuning or downstream transferability. He et al. (2022) further argues that linear probing misses the opportunity to utilize strong but non-linear features, and this sentiment is repeated by a number of follow-up research efforts (Yi et al., 2022; Chen et al., 2022a). The most recent research effort on this topic is the work of (Park et al., 2023) where the authors argue that models trained with MIM exhibit a bias towards texture whereas contrastive learning leads to a bias towards shape, suggesting that this may be the explanation for the difference in linear probing accuracy. Nevertheless, thorough investigations on SSL model evaluations and convincing explanations for their differences are largely missing.

**On the usage of tokenizers in SSL** – From Table 8 it can be seen that a number of MIM-based generative SSL frameworks make use of a previously trained tokenizer in order to enhance learned representations. In particular, (Peng et al., 2022; Gao et al., 2023; Zhang et al., 2022d) demonstrate that the usage of `CLIP` tokenizers (either as-is or as a teacher for distillation) boosts the performance of frameworks considerably. However, using a tokenizer that is trained on a large corpus of images to demonstrate state-of-the-art results against other frameworks that do not have access to this level of supervision has recently attracted criticism from the community (ICLR, 2023). We believe that the investigation of the efficacy and necessity of tokenizers in generative SSL is an area that is largely unexplored.

**Moving forward: generative or discriminative SSL?** – Although recent results obtained with generative SSL seem to favor generative approaches over discriminative ones, it is important to note that generative SSL has not only benefited from the discovery of vision transformers, but also from advancements made in the area of discriminative SSL. The most recent comparative studies suggest that the answer to the discriminative versus generative question is not that straightforward and that both approaches have their own merits and limitations (Park et al., 2023). As mentioned, some of the newly proposed generative frameworks also leverage discriminative features such as contrastive learning (Yi et al., 2022; Zhang et al., 2022d) or distillation (Zhou et al., 2021; Peng et al., 2022). Therefore, we speculate that this trend will continue and that the newly proposed frameworks in the upcoming years will leverage improvements from both sides in order to further improve the state-of-the-art.

**Cadence of research in SSL and the extent of this survey** – Before we conclude our survey, we would like to briefly discuss the cadence of research in SSL and the breadth of topics covered in this survey.

It is widely recognized that the field of machine learning has experienced an unprecedented growth in research and development over the past decade, particularly following the groundbreaking results achieved with AlexNet (Krizhevsky et al., 2012). During this period, significant advances have been made not only in the architectural design of deep neural networks, but also in optimizers, training routines, normalization techniques, data augmentation, and various other areas. While these improvements have steadily advanced the state-of-the-art in supervised learning, self-supervised learning has also benefited from the majority of these advancements from the beginning. Thanks to the integration of these advancements into SSL from its early stages, as well as the availability of computational resources, the state-of-the-art in SSL has improved rapidly since 2018. The pace of research has been so fast that some frameworks have been improved upon even before their predecessors got published (e.g., `CAE` (Chen et al., 2022a) and `CAE-v2` (Zhang et al., 2022d), `BEiT-v2` (Peng et al., 2022) and `BEiT-v3` (Wang et al., 2022a)).

Given the aforementioned observation, to cover the most up-to-date research efforts, we have decided to include papers that have not yet been published in conference proceedings or journals. Consequently, the majority of the papers cited in this survey, from 2021 onward, are preprints. At the time of submitting this survey, the latest conference surveyed for relevant papers was the 11th International Conference on Learning Representations (ICLR), held in May 2023, from which a portion of the papers cited in this survey have been rejected, withdrawn, or published (Wang et al., 2023; Fang et al., 2023; Park et al., 2023; Chen et al., 2022a; Peng et al., 2022; Chen et al., 2023b; Gao et al., 2023; Mishra et al., 2023; Li et al., 2023; Kukleva et al., 2023; Zhao et al., 2023; Chen et al., 2023c; Park et al., 2023).

## Funding Information

This work was supported by a grant from the Special Research Fund (BOF) of Ghent University (BOF/STA/202109/039).

# Appendix for:
# Know Your Self-supervised Learning:
# A Survey on Image-based Generative and Discriminative Training

The content of appendix is detailed below.

- **A list of abbreviations** is provided in Section A.

- **Metadata** for the frameworks such as the primary affiliation of authors, publication date, and source code as well as availability of trained models are provided for:
    - Discriminative SSL frameworks in Table 17
    - Enhancements to existing SSL frameworks in Table 18
    - Generative SSL frameworks in Table 19

- **Repositories** that are useful for vision-based SSL are listed in Table 20.

- **Datasets** used for the evaluation in the respective papers of the frameworks are provided for:
    - Discriminative SSL frameworks in Table 21
    - Enhancements to discriminative SSL frameworks in Table 22
    - Generative SSL frameworks in Table 23

- **Benchmarks** on ImageNet-1K are provided for:
    - Clustering-based SSL frameworks in Table 24
    - Contrastive-learning-based SSL frameworks in Table 25
    - Distillation-based SSL frameworks in Table 26
    - Information-maximization-based SSL frameworks in Table 27
    - Enhancements to discriminative SSL frameworks in Table 28
    - `GAN`-based SSL frameworks in Table 29
    - MIM-based SSL frameworks in Table 30

- **Benchmarks** on COCO for all frameworks are provided in Table 31.

# A   List of abbreviations

Clustering frameworks

| | |
|---|---|
| SeLa | Self labeling |
| SCAN | Semantic clustering by adopting nearest neighbors |
| Swav | Swapping assignments between multiple views of the same image |
| ODC | Online deep clustering |
| CoKe | Constrained K-means |

Contrastive frameworks

| | |
|---|---|
| InstDist | Instance discrimination |
| NPID | Non-parametric instance discrimination |
| CPC | Contrastive predictive coding |
| DIM | Deep InfoMax |
| AMDIM | Augmented multi-scale DIM |
| CMC | Contrastive multi-view coding |
| MoCo | Momentum contrast |
| PIRL | Pretext-invariant representation learning |
| SimCLR | A simple framework for contrastive learning |
| PCL | Prototypical contrastive learning |
| PIC | Parametric instance classification |
| DCL | Debiased contrastive learning |
| LooC | Leave-one-out contrastive learning |
| G-SimCLR | Self-supervised contrastive learning with guided projection |
| ReLIC | Representation learning via invariant causal mechanisms |
| AdCo | Adversarial contrast |
| DenseCL | Dense contrastive learning |
| PixPro | Pixel-level consistency propagation |
| CLSA | Contrastive learning with stronger augmentations |
| NNCLR | Nearest-neighbor contrastive learning of visual representations |
| MoBY | MoCo + BYOL |
| DNC | Divide and contrast |
| ReSSL | Relational self-supervised learning |
| UniGrad | A unified gradient framework |
| SimCo | Simplified MoCo without momentum |
| SimMoCo | Simplified MoCo |
| UniVIP | A unified framework for self-supervised visual pre-training |
| Mugs | Multi-granular self-supervised learning |
| CaCo | Cooperative-adversarial contrastive learning |
| SMoG | Synchronous momentum grouping |
| SiameseIM | Siamese image modelling |

Distillation frameworks

| | |
|---|---|
| BYOL | Build your-own latent |
| SimSiam | Simple Siamese representation learning networks |
| OBoW | Online bag-of-visual-words |
| DirectPred | Direct linear predictor |
| SEED | Self-supervised distillation for visual representation |
| DisCo | Distilled contrastive learning |
| DINO | Self-distillation with no labels |
| EsVit | Efficient self-supervised vision transformer |
| BINGO | Bag of instances aggregation |
| TinyMIM | Tiny MIM |

Information-maximization frameworks

| | |
|---|---|
| WMSE | Whitening mean squared error |
| VicReg | Variance-invariance-covariance regularization |
| TWIST | Twin class distribution estimation |
| TLDR | Twin learning for dimensionality reduction |
| ARB | Align representations with base |
| VicRegL | VicReg with local visual features |

Enhancement modules

| | |
|---|---|
| InfoMin | Mutual information principle |
| InterCLR | Inter-image contrastive learning |
| HEXA | Hard examples |
| MocHi | Mixing of contrastive hard negatives |
| Resim | Region similarity representation learning |
| MSF | Mean shift for self-supervised learning |
| ORL | Object-level representation learning |
| CEB | Conditional entropy bottleneck |
| SEM | Simplicial embeddings |
| ENS | Ensemble self-supervised learning |
| MRCL | Masked reconstruction contrastive learning |
| TS | Temperature schedules |
| ARCL | Augmentation-robust contrastive learning |
| MosRep | Mosaic representation learning framework |

GAN-based frameworks

| | |
|---|---|
| BiGAN | Bidirectional GAN |
| ALI | Adversarially learned inference |
| BigBiGAN | BiGAN with BigGAN generator |
| SS-GAN | Self-supervised GAN |
| SS-GAN-LA | SS-GAN with label augmentation |
| VQGAN | Vector-quantized GAN |
| Vit-VQGAN | ViT-based VQGAN |

MIM-based frameworks

| | |
|---|---|
| iGPT | Image GPT |
| BEiT | Bidirectional encoder representation from image transformers |
| MAE | Masked autoencoders |
| iBOT | Image BERT pre-training with online tokenizer |
| SimMIM | A simple framework for MIM |
| PeCO | Perceptual codebook |
| MaskFeat | Masked feature prediction |
| CAE | Context autoencoder |
| CIM | Corrupted image modeling |
| MCMAE | Masked convolution meets MAE |
| ConMIM | Denoising contrast masked image modeling |
| CMAE | Contrastive MAE |
| SdAE | Self-distilled masked autoencoder |
| MILAN | Masked image pretraining on language assisted representation |
| CAN | Contrastive learning, masked autoencoders, and noise prediction |
| PCAE | Progressively compressed autoencoder |
| SparK | Sparse masked modeling |
| MRMAE | Mimic before reconstruct MAE |

Others

| | |
|---|---|
| BERT | Bidirectional encoder representations from transformers |
| GPT | Generative pre-trained transformer |

## B   Metadata for frameworks

| SSL framework | Primary affiliation | Publication date | Experiments on on ImageNet 1K | Downstream experiments | Official implementation | Trained models |
|---|---|---|---|---|---|---|
| Deep Cluster | Facebook AI Research | Mar 2019 | Yes | Yes | Available | Available |
| Local Aggregation | Stanford University | Apr 2019 | Yes | Yes | Available | Not available |
| Deeper Cluster | Facebook AI Research | Aug 2019 | Yes | Yes | Available | Available |
| SeLa | University of Oxford | Nov 2019 | Yes | Yes | Available | Available |
| SCAN | KU Leuven | Jul 2020 | Yes | No | Available | Available |
| Deep Cluster-v2 | Facebook AI Research | Jun 2020 | Yes | Yes | Available | Available |
| SeLa-v2 | Facebook AI Research | Jun 2020 | Yes | Yes | Available | Available |
| Swav | Facebook AI Research | Jun 2020 | Yes | Yes | Available | Available |
| ODC | SenseTime | Jun 2020 | Yes | Yes | Available | Not available |
| CoKe | Alibaba | May 2021 | Yes | Yes | Available | Available |
| Self-Classifier | IBM Research | Jul 2022 | Yes | Yes | Available | Available |
| InstDist (NPID) | Chinese Univ. of Hong Kong | May 2018 | Yes | Yes | Available | Available |
| CPC | DeepMind | Jul 2018 | Yes | No | Not available | Not available |
| DIM | Microsoft Research | Aug 2018 | No | No | Available | Not available |
| CPC-v2 | DeepMind | May 2019 | Yes | Yes | Not available | Not available |
| AMDIM | Microsoft Research | Jun 2019 | Yes | No | Available | Available |
| CMC | MIT | Jun 2019 | Yes | Yes | Available | Available |
| MoCo | Facebook AI Research | Nov 2019 | Yes | Yes | Available | Available |
| PIRL | Facebook AI Research | Dec 2019 | Yes | Yes | Not available | Not available |
| SimCLR | Google Research | Feb 2020 | Yes | Yes | Available | Available |
| MoCo-v2 | Facebook AI Research | Mar 2020 | Yes | Yes | Available | Available |
| SimCLR-v2 | Google Research | Jun 2020 | Yes | No | Available | Available |
| PCL & PCLv2 | Salesforce Research | Jun 2020 | Yes | Yes | Available | Available |
| PIC | Microsoft Research | Jun 2020 | Yes | Yes | Available | Available |
| DCL | MIT | Jul 2020 | No | Yes | Available | Available |
| LooC | UC Berkeley | Aug 2020 | Yes | Yes | Not available | Not available |
| G-SimCLR | Walmart Labs | Sep 2020 | No | No | Available | Available |
| ReLIC | DeepMind | Oct 2020 | Yes | Yes | Not available | Not available |
| AdCo | Peking University | Nov 2020 | Yes | Yes | Available | Available |
| DenseCL | The University of Adelaide | Nov 2020 | No | Yes | Available | Available |
| PixPro | Microsoft Research | Nov 2020 | Yes | Yes | Available | Available |
| MoCo-v3 | Facebook AI Research | Apr 2021 | Yes | Yes | Available | Available |
| CLSA | Purdue University | Apr 2021 | Yes | Yes | Available | Available |
| Truncated Triplet | Sun Yat-sen University | Apr 2021 | Yes | Yes | Available | Available |
| NNCLR | Google Research | Apr 2021 | Yes | Yes | Not available | Not available |
| MoBY | Microsoft Research | May 2021 | Yes | Yes | Available | Available |
| DNC | DeepMind | May 2021 | Yes | Yes | Not available | Not available |
| ReSSL | SenseTime | Jul 2021 | Yes | No | Available | Available |
| UniGrad | SenseTime | Dec 2021 | Yes | No | Available | Available |
| ReLIC-v2 | DeepMind | Jan 2022 | Yes | Yes | Not available | Not available |
| SimCo | KAIST | Mar 2022 | No | No | Available | Not available |
| SimMoCo | KAIST | Mar 2022 | No | No | Available | Not available |
| UniVIP | University of Chinese AoS | Mar 2022 | Yes | Yes | Not available | Not available |
| Mugs | Sea AI Lab | Mar 2022 | Yes | Yes | Available | Available |
| CaCo | Purdue University | Mar 2022 | Yes | Yes | Available | Available |
| SMoG | Huawei | Jul 2022 | Yes | Yes | Not available | Not available |
| SiameseIM | Shanghai Artificial Intelligence Lab. | Nov 2022 | Yes | Yes | Not available | Not available |
| BYOL | DeepMind | Jun 2020 | Yes | Yes | Available | Available |
| SimSiam | Facebook AI Research | Aug 2020 | Yes | Yes | Available | Available |
| OBoW | Valeo.ai | Dec 2020 | Yes | Yes | Available | Available |
| SEED | Microsoft Research | Jan 2021 | Yes | Yes | Not available | Not available |
| DirectPred | Facebook AI Research | Feb 2021 | Yes | Yes | Available | Not available |
| DisCO | Tencent | Apr 2021 | Yes | Yes | Available | Available |
| DINO | Facebook AI Research | Apr 2021 | Yes | Yes | Available | Available |
| EsViT | Microsoft Research | Jun 2021 | Yes | Yes | Available | Available |
| BINGO | Huawei | Mar 2022 | Yes | Yes | Available | Not available |
| TinyMIM | Microsoft Research | Jan 2023 | Yes | Yes | Available | Available |
| WMSE | University of Trento | Jul 2020 | Yes | No | Available | Not available |
| Barlow Twins | Facebook AI Research | Mar 2021 | Yes | Yes | Available | Available |
| VicReg | Facebook AI Research | May 2021 | Yes | Yes | Available | Available |
| TWIST | Tsinghua University | Oct 2021 | Yes | Yes | Available | Available |
| TLDR | Naver Labs EU | Oct 2021 | Yes | Yes | Available | Not available |
| ARB | Shanghai Jiao Tong University | Nov 2021 | Yes | No | Not available | Not available |
| VicRegL | Facebook AI Research | Oct 2022 | Yes | Yes | Available | Available |

Table 17: Publication information as well as implementation details for discriminative SSL frameworks covered in this survey.

| SSL framework | Primary affiliation | Publication date | Experiments on on ImageNet 1K | Downstream experiments | Official implementation | Trained models |
|---|---|---|---|---|---|---|
| InfoMin | MIT | May 2020 | Yes | Yes | Available | Available |
| InterCLR | Nanyang Technological Univ. | Aug 2020 | Yes | Yes | Not available | Not available |
| HEXA | Microsoft Research | Dec 2020 | Yes | Yes | Not available | Not available |
| MocHi | Naver Labs EU | Oct 2020 | Yes | Yes | Not available | Available |
| ReSim | UC Berkeley | Mar 2021 | Yes | Yes | Available | Available |
| MSF | University of Maryland | May 2021 | Yes | Yes | Available | Available |
| ORL | Nanyang Technological Univ. | Jun 2021 | Yes | Yes | Available | Available |
| CEB | Google Research | Sep 2021 | Yes | Yes | Available | Available |
| SEM | MILA | Apr 2022 | Yes | Yes | Available | Not available |
| ENS | Google Research | Nov 2022 | Yes | Yes | Not available | Not available |
| MRCL | University of Chinese AoS | Nov 2022 | Yes | Yes | Not available | Not available |
| TS | Max Planck Institute | Mar 2023 | No | No | Available | Available |
| ARCL | Shanghai Jiao Tong Univ. | Mar 2023 | Yes | Yes | Not available | Not available |
| MosRep | University of Sydney | Mar 2023 | Yes | Yes | Available | Available |

Table 18: Publication information as well as implementation details for enhancements proposed to existing SSL frameworks covered in this survey.

| SSL framework | Primary affiliation | Publication date | Experiments on on ImageNet 1K | Downstream experiments | Official implementation | Trained models |
|---|---|---|---|---|---|---|
| BiGAN | University of California | May 2016 | Yes | Yes | Available | Available |
| BigBiGAN | DeepMind | Jul 2019 | Yes | Yes | Available | Available |
| ALI | MILA | Jun 2016 | No | Yes | Available | Available |
| SS-GAN | University of California | Nov 2018 | Yes | Yes | Available | Not available |
| SS-GAN-LA | University of Chinese AoS | Oct 2021 | No | Yes | Available | Available |
| Vit-VQGAN | Google Research | Oct 2021 | Yes | Yes | Not available | Not available |
| iGPT | OpenAI | Jul 2020 | Yes | Yes | Available | Available |
| BEiT | Microsoft Research | Jun 2021 | Yes | Yes | Available | Available |
| MAE | Facebook AI Research | Nov 2021 | Yes | Yes | Available | Available |
| iBOT | ByteDance | Nov 2021 | Yes | Yes | Available | Available |
| SimMIM | Microsoft Research | Nov 2021 | Yes | Yes | Available | Available |
| PeCO | Microsoft Research | Nov 2021 | Yes | Yes | Not available | Not available |
| MaskFeat | Facebook AI Research | Dec 2021 | Yes | Yes | Available | Available |
| data2vec | Facebook AI Research | Feb 2022 | Yes | No | Available | Available |
| CAE | Peking University | Feb 2022 | Yes | Yes | Not available | Not available |
| CIM | Microsoft Research | Feb 2022 | Yes | Yes | Not available | Not available |
| MCMAE | SenseTime | May 2022 | Yes | Yes | Available | Available |
| ConMIM | Tencent | May 2022 | Yes | Yes | Available | Available |
| CMAE | ByteDance | Jul 2022 | Yes | Yes | Not available | Not available |
| SdAE | Huawei | Jul 2022 | Yes | Yes | Available | Available |
| MILAN | Alibaba | Aug 2022 | Yes | Yes | Available | Available |
| BEiT-v2 | Microsoft Research | Aug 2022 | Yes | Yes | Available | Available |
| BEiT-v3 | Microsoft Research | Aug 2022 | Yes | Yes | Available | Available |
| CAE-v2 | Baidu | Nov 2022 | Yes | Yes | Not available | Not available |
| CAN | Google Research | Jan 2023 | Yes | Yes | Available | Available |
| PCAE | Huawei | Jan 2023 | Yes | Yes | Available | Available |
| SparK | ByteDance | Jan 2023 | Yes | Yes | Available | Available |
| MRMAE | Shanghai AI Laboratory | Mar 2023 | Yes | Yes | Available | Available |

Table 19: Publication information as well as implementation details for generative SSL frameworks covered in this survey.

| Repository name | Maintainer | Purpose |
|---|---|---|
| Awesome SSL | Independent | A comprehensive reading list for SSL |
| solo-learn | Independent (da Costa et al., 2022) | SSL frameworks, benchmarking, and model zoo |
| VISSL | Facebook (Goyal et al., 2021b) | SSL frameworks, benchmarking, and model zoo |
| MMSelfSup | OpenMMLab (Contributors, 2021) | SSL frameworks, benchmarking, and model zoo |
| Lightly | Lightly.ai (Susmelj et al., 2020) | SSL frameworks and benchmarking |
| EasyCV | Alibaba (Contributors, 2022) | SSL frameworks and benchmarking |
| Unified SSL Benchmark | Microsoft (Wang et al., 2022c) | SSL frameworks and benchmarking |

Table 20: Github repositories related to SSL, their maintainer, and purpose.

## C  Framework dataset usage

| SSL framework | Used datasets | Tasks |
|---|---|---|
| Deep Cluster | ImageNet-1k, Pascal VOC, Places, YFCC100M | C, D, S |
| Local Aggregation | ImageNet-1k, Pascal VOC | C, D |
| Deeper Cluster | ImageNet-1k, Pascal VOC, Places | C, D, S |
| SeLa | ImageNet-1k, CIFAR-100, CIFAR-10, Pascal VOC, SVHN | C, D |
| SCAN | ImageNet-1k, CIFAR-100, CIFAR-10, STL-10 | C |
| Deep Cluster-v2 | ImageNet-1k | C |
| SeLa-v2 | ImageNet-1k | C |
| Swav | ImageNet-1k, COCO, Pascal VOC, Places | C, D |
| ODC | ImageNet-1k, Pascal VOC, Places | C, D |
| CoKe | ImageNet-1k, COCO, Pascal VOC | C, D, S |
| Self-Classifier | ImageNet-1k, COCO, Pascal VOC | C, D |
| InstDist (NPID) | ImageNet-1k, Places | C, D |
| CPC | ImageNet-1k | C |
| DIM | Tiny ImageNet, CIFAR-100, CIFAR-10, STL-10, CelebA | C |
| CPC-v2 | ImageNet-1k, Pascal VOC | C, D |
| AMDIM | ImageNet-1k, CIFAR-100, CIFAR-10, Places, STL-10 | C |
| CMC | ImageNet-1k, STL-10 | C, D, S |
| MoCo | ImageNet-1k, COCO, Pascal VOC | C, D, S |
| PIRL | ImageNet-1k, Pascal VOC, Places, iNat | C, D |
| SimCLR | ImageNet-1k, CIFAR-100, CIFAR-10, Pascal VOC, Food, Birdsnap SUN397, Cars, Aircraft, DTD, Pets, Caltech-101, Flower | C, D |
| MoCo-v2 | ImageNet-1k, Pascal VOC | C, D |
| SimCLR-v2 | ImageNet-1k, CIFAR-10 | C |
| PCL & PCLv2 | ImageNet-1k, Pascal VOC, Places | C, D |
| PIC | ImageNet-1k, Pascal VOC, Cityscapes, iNat18 | C, D, S |
| DCL | ImageNet-100, CIFAR-10, STL-10 | C |
| LooC | ImageNet-100, iNat-1K, CUB-200, Flowers-102 | C |
| G-SimCLR | ImageNet subset, CIFAR-10 | C |
| ReLIC | ImageNet-1k, ImageNet-R, ImageNet-C | C |
| AdCo | ImageNet-1k, COCO, Pascal VOC, Places | C, D |
| DenseCL | COCO, Pascal VOC, Cityscapes | C, D, S |
| PixPro | ImageNet-1k, COCO, Pascal VOC, Cityscapes | C, D, S |
| MoCo-v3 | ImageNet-1k, CIFAR-100, CIFAR-10, Oxford Flowers-102 , Oxford-IIIT-Pet | C |
| CLSA | ImageNet-1k, COCO, Pascal VOC | C, D |
| Truncated Triplet | ImageNet-1k, COCO, Pascal VOC, SYSU-30k | C, D, S |
| NNCLR | ImageNet-1k, CIFAR-100, CIFAR-10, Pascal VOC, Food, Birdsnap SUN397, Cars, Aircraft, DTD, Pets, Caltech-101, Flower | C, D |
| MoBY | ImageNet-1k, COCO, ADE20K | C, D, S |
| DNC | ImageNet-1k, CIFAR-100, CIFAR-10, COCO, Pascal VOC, Places, Food Birdsnap, SUN397, Cars, Aircraft, DTD, Pets, Caltech-101, Flower, NYU v2 | C, D, S |
| ReSSL | ImageNet-1k, CIFAR-100, CIFAR-10, STL-10 | C |
| UniGrad | ImageNet-1k | C |
| ReLIC-v2 | ImageNet-1k, ImageNetV2, ImageNet-C, ImageNet-R ImageNet-Sketch, PASCAL VOC, Cityscapes | C, D, S |
| SimCo | ImageNet-100, CIFAR-100, CIFAR-10, STL-10, SVHN | C |
| SimMoCo | ImageNet-100, CIFAR-100, CIFAR-10, STL-10, SVHN | C |
| UniVIP | ImageNet-1k, COCO | C, D, S |
| Mugs | ImageNet-1k, COCO | C, D, S |
| CaCo | ImageNet-1k, CIFAR-100, CIFAR-10, Pascal VOC, Food SUN397, Cars, Aircraft, DTD, Pets, Caltech-101, Flower | C, D |
| SMoG | ImageNet-1k, COCO, Pascal VOC, Cityscapes | C, D, S |
| SiameseIM | ImageNet-1k, COCO, ADE20k | C, D, S |
| BYOL | ImageNet-1k, CIFAR-100, CIFAR-10, Pascal VOC, Food Birdsnap, SUN397, Cars, Aircraft, DTD, Pets, Caltech-101, Flower, NYU v2 | C, D, S |
| SimSiam | ImageNet-1k, COCO, Pascal VOC | C, D, S |
| OBoW | ImageNet-1k, Pascal VOC, Places | C, D, S |
| DirectPred | ImageNet-1k, CIFAR-10, STL-10 | C |
| SEED | ImageNet-1k, CIFAR-10, STL-10, COCO, Pascal VOC | C, D, S |
| DisCO | ImageNet-1k, CIFAR-100, CIFAR-10, COCO, Pascal VOC | C, D, S |
| DINO | ImageNet-1k, CIFAR-100, CIFAR-10, Pascal VOC, iNat18, iNat19 Flowers, Cars, iNet, Google Landmarks v2, DAVIS 2017 videos | C, D, S |
| EsViT | ImageNet-1k, CIFAR-100, CIFAR-10, COCO, Pascal VOC, STL-10, MINST, Food SUN397, Cars, Aircraft, DTD, Pets, Caltech-101, Flower, FER2013, GTSRB, HatefulMemes, PatchCamelyon, UCF101 | C, D, S |
| BINGO | ImageNet, CIFAR-100, CIFAR-10, COCO | C, D, S |
| TinyMIM | ImageNet-1k, ADE20K | C, S |
| WMSE | ImageNet-1k, ImageNet-100, Tiny ImageNet, CIFAR-100, CIFAR-10, STL-10 | C |
| Barlow Twins | ImageNet-1k, COCO, Pascal-VOC, Places, iNat18 | C, D, S |
| VicReg | ImageNet-1k,COCO, Pascal VOC, Places, iNat18 | C, D, S |
| TWIST | ImageNet-1k, CIFAR-100, CIFAR-10, COCO, Pascal VOC, Food SUN397, Cars, Aircraft, DTD, Pets, Caltech-101, Flower | C, D, S |
| TLDR | ImageNet-1k | C |
| ARB | ImageNet-1k, ImageNet-100, CIFAR-100, CIFAR-10 | C |
| VicRegL | ImageNet-1k, Pascal VOC, Cityscapes | C, S |

Table 21: Employed datasets for experiments used in the research articles of the discriminative SSL frameworks. The third column summarizes the evaluated tasks in the respective papers of frameworks: (C)lassification, (D)etection/localization, and (S)egmentation.

| SSL framework | Used datasets | Tasks |
|---|---|---|
| InfoMin | ImageNet-1k, COCO, Pascal VOC, Colorful Moving-MNIST | C, D, S |
| InterCLR | ImageNet-1k, Pascal VOC, Places | C, D |
| HEXA | ImageNet-1k, CIFAR-100, CIFAR-10, Pascal VOC | C, D |
| MocHi | ImageNet-1k, ImageNet-100, COCO, Pascal VOC | C, D, S |
| ReSim | ImageNet-1k, ImageNet-100, COCO, Pascal VOC | D, S |
| MSF | ImageNet-1k, CIFAR-100, CIFAR-10, Pascal VOC, Food, SUN397, Cars Aircraft, DTD, Pets, Caltech-101, Flower | C, D |
| ORL | ImageNet-1k, COCO, Pascal VOC, Places, iNat | C, D, S |
| CEB | ImageNet-1k, ImageNet-A, ImageNet-C, ImageNet-R ImageNet-v2, ImageNet-Vid, YouTube-BB, ObjectNet | C |
| SEM | ImageNet-1k, ImageNet-A, ImageNet-C, ImageNet-R ImageNet-v2, CIFAR-100, CIFAR-10, Food, Sun397, DTD, Flower | C |
| ENS | ImageNet-1k, CIFAR-100, CIFAR-10, Food, SUN397, Cars DTD, Pets, Caltech-101, Flower | C |
| MRCL | ImageNet-1k, COCO, ADE20k | C, D, S |
| TS | ImageNet-100, CIFAR-100, CIFAR-10 | C |
| ARCL | ImageNet-1k, CIFAR-100, CIFAR-10, Food, SUN397 Cars, Aircraft, DTD, Pets, Caltech-101, Flower | C |
| MosRep | ImageNet-1k, ImageNet-100, CIFAR-100, CIFAR-10, COCO Food, Cars, DTD, Pets, Caltech-101, Flower | C, D, S |

Table 22: Employed datasets for experiments used in the research articles of the enhancements to discriminative SSL frameworks. The third column summarizes the evaluated tasks in the respective papers of frameworks: (C)lassification, (D)etection/localization, and (S)egmentation.

| SSL framework | Used datasets | Tasks |
|---|---|---|
| BigBiGAN | ImageNet-1k | C |
| BiGAN | ImageNet-1k, Pascal VOC, MNIST | C, D, S |
| ALI | Tiny ImageNet, CIFAR-10, SVHN, CelebA | C |
| SS-GAN | ImageNet-1k, CIFAR-10, CelebA-HQ, LSUN-Bedroom | C |
| SS-GAN-LA | Tiny-ImageNet, CIFAR-10, STL-10, CelebA | C |
| Vit-VQGAN | ImageNet-1k, CelebA-HQ, FFHQ | C |
| iGPT | ImageNet-1k, CIFAR-100, CIFAR-10, STL-10 | C |
| BEiT | ImageNet-1k, ADE20K | C, S |
| MAE | ImageNet-1k, COCO, Places, iNat17, iNat18, iNat19 | C, D, S |
| iBOT | ImageNet-1k, CIFAR-100, CIFAR-10, COCO, ADE20k, iNat18, iNat19, Flower, Cars | C, D, S |
| SimMIM | ImageNet-1k, COCO, ADE20K, iNat18 | C, D, S |
| PeCO | ImageNet-1k, COCO, ADE20k | C, D, S |
| MaskFeat | ImageNet-1k, Kinetics-400, Kinetics-600, Kinetics-700 | C |
| data2vec | ImageNet-1k | C |
| CAE | ImageNet-1k, COCO, ADE20K | C, D, S |
| CIM | ImageNet-1k, COCO, ADE20K | C, D, S |
| MCMAE | ImageNet-1k, COCO, ADE20K | C, D, S |
| ConMIM | ImageNet-1k, COCO, ADE20K | C, D, S |
| CMAE | ImageNet-1k, COCO, ADE20K | C, D, S |
| SdAE | ImageNet-1k, COCO, ADE20K | C, D, S |
| MILAN | ImageNet-1k, COCO, ADE20K | C, D, S |
| BEiT-v2 | ImageNet-1k, ADE20K | C, S |
| BEiT-v3 | ImageNet-1k, COCO, ADE20K | C, D, S |
| MRCL | ImageNet-1k, COCO, ADE20K | C, D, S |
| CAE-v2 | ImageNet-1k, COCO, ADE20K | C, D, S |
| CAN | ImageNet-1k, ImageNet-v2, ImageNet-Real, ImageNet-Adversarial, ImageNet-Rendition CIFAR-100, Birds, Cars, DTD, Pets, UC-Merced, Col-Hist, Caltech, ObjectNet | C |
| PCAE | ImageNet-1k, COCO | C, D, S |
| SparK | ImageNet-1k, COCO | C, D, S |
| MRMAE | ImageNet-1k, COCO | C, D |

Table 23: Employed datasets for experiments used in the research articles of the generative SSL frameworks. The third column summarizes the evaluated tasks in the respective papers of frameworks: (C)lassification, (D)etection/localization, and (S)egmentation.

# D ImageNet benchmarks

| SSL framework | Backbone network | SSL epochs | Fine-tuning accuracy | Linear probing accuracy | Additional notes |
|---|---|---|---|---|---|
| Deep Cluster | AlexNet | 500 | - | 39.8 | Used conv4 output |
| LA | ResNet-50 | 200 | - | 60.2 | - |
| Deeper Cluster | VGG-16 | 100 | - | 48.4 | - |
| SeLA | ResNet-50 | 90 | - | 61.5 | - |
| SCAN | ResNet-50 | 90 | - | 39.9 | Unsupervised evaluation |
| Deep Cluster-v2 | ResNet-50 | 400 | - | 74.3 | 2x160 + 4x96 crops |
| Deep Cluster-v2 | ResNet-50 | 800 | 71.9 | - | - |
| SeLA-v2 | ResNet-50 | 400 | - | 71.8 | 2x160 + 4x96 crops |
| Swav | ResNet-50 | 800 | 77.8 | 75.3 | - |
| ODC | ResNet-50 | 440 | - | 57.6 | - |
| CoKe | ResNet-50 | 800 | - | 76.4 | 8 views |
| Self-C. | ResNet-50 | 800 | - | 74.1 | - |

Table 24: ImageNet-1k linear probing and fine-tuning benchmarks for **clustering**-based SSL frameworks.

| SSL framework | Backbone network | SSL epochs | Fine-tuning accuracy | Linear probing accuracy | Additional notes |
|---|---|---|---|---|---|
| InstDist (NPID) | ResNet-50 | 200 | - | 54.0 | Using conv5 |
| CPC | ResNet-v2 101 | 130 | - | 48.7 | - |
| CPCv2 | ResNet-50 | 200 | - | 61.8 | - |
| AMDIM | ResNet-50 | 150 | - | 68.1 | Large AMDIM model |
| CMC | ResNet-50 | 200 | - | 66.2 | RandAugment |
| MoCo | ResNet-50 | 200 | 77.3 | 60.6 | - |
| MoCo | ResNet50 (×2) | 200 | - | 65.4 | - |
| MoCo | ResNet50 (×4) | 200 | - | 68.6 | - |
| PIRL | ResNet-50 | 800 | - | 63.6 | At res5 |
| SimCLR | ResNet-50 | 100 | - | 63.6 | - |
| SimCLR | ResNet-50 (×2) | 100 | - | 74.2 | - |
| SimCLR | ResNet-50 (×4) | 100 | - | 76.5 | - |
| MoCo-v2 | ResNet-50 | 800 | 75.5 | 71.1 | - |
| SimCLR-v2 | ResNet-50 | 400 | 76.3 | 71.7 | - |
| SimCLR-v2 | ResNet-50 (×2) | 400 | 79.1 | 75.6 | - |
| SimCLR-v2 | ResNet-101 | 400 | 78.2 | 73.6 | - |
| SimCLR-v2 | ResNet-101 (×2) | 400 | 80.7 | 77.0 | - |
| PCL | ResNet-50 | 200 | - | 61.5 | - |
| PCL-v2 | ResNet-50 | 200 | - | 67.6 | - |
| PIC | ResNet-50 | 200 | - | 70.8 | - |
| ReLIC | ResNet-50 | 800 | - | 74.8 | - |
| AdCo | ResNet-50 | 800 | - | 75.7 | Multi-crop |
| AdCo | ResNet-50 | 200 | 67.0 | - | - |
| PixPro | ResNet-50 | 100 | - | 66.3 | with SimCLR |
| MoCo-v3 | ResNet-50 | 800 | - | 73.8 | - |
| MoCo-v3 | ViT-B | 300 | 83.2 | 76.7 | - |
| MoCo-v3 | ViT-L | 300 | 84.1 | 77.6 | - |
| MoCo-v3 | ViT-H | 300 | - | 78.1 | - |
| CLSA | ResNet-50 | 200 | - | 73.3 | Multi-crop |
| CLSA | ResNet-50 | 800 | - | 76.2 | Multi-crop |
| Truncated Triplet | ResNet-50 | 700 | - | 75.9 | - |
| NNCLR | ResNet-50 | 1000 | - | 75.6 | Multi-crop |
| MoBY | DeiT-S | 300 | - | 72.8 | - |
| MoBY | Swin-T | 300 | - | 75.0 | - |
| DNC | ResNet-50 | 3000 | 78.2 | 75.8 | - |
| ReSSL | ResNet-50 | 200 | - | 74.7 | 5 crops |
| UniGrad | ResNet-50 | 800 | - | 75.5 | with CutMix + multi-crop |
| ReLIC-v2 | ResNet-50 | 1000 | - | 77.1 | - |
| UniVIP | ResNet-50 | 300 | - | 74.2 | - |
| Mugs | ViT-S | 3200 | 82.6 | 78.9 | - |
| Mugs | ViT-B | 1600 | 84.3 | 80.6 | 84.3 |
| Mugs | ViT-L | 1000 | - | 82.1 | - |
| CaCo | ResNet-50 | 200 | - | 75.3 | - |
| SMoG | ResNet-50 | 400 | 78.3 | 76.4 | Multi-crop |
| SiameseIm | ViT-B | 1600 | 84.1 | 78.0 | - |

Table 25: ImageNet-1k linear probing and fine-tuning benchmarks for **contrastive-learning**-based SSL frameworks.

| SSL framework | Backbone network | SSL epochs | Fine-tuning accuaracy | Linear probing accuracy | Additional notes |
|---|---|---|---|---|---|
| BYOL | ResNet-50 | 1000 | 77.7 | 74.3 | - |
| BYOL | ResNet-50 ($\times$2) | 1000 | - | 77.4 | - |
| BYOL | ResNet-50 ($\times$4) | 1000 | - | 78.6 | - |
| BYOL | ResNet-200 ($\times$2) | 1000 | - | 79.6 | - |
| SimSiam | ResNet-50 | 800 | - | 71.3 | - |
| OBoW | ResNet-50 | 200 | - | 73.8 | - |
| DirectPred | ResNet-50 | 300 | - | 72.4 | - |
| SEED | ResNet-34 | 800 | - | 58.5 | Distilled from ResNet-50 |
| DisCo | ResNet-34 | 800 | - | 62.5 | Distilled from ResNet-50 |
| DINO | ResNet-50 | 300 | - | 75.3 | - |
| DINO | ViT-S | 300 | 81.5 | 77.0 | - |
| DINO | ViT-B | 300 | 82.8 | 78.2 | - |
| EsViT | Swin-T | 300 | - | 78.1 | - |
| EsViT | Swin-S | 300 | - | 79.5 | - |
| EsViT | Swin-B | 300 | - | 80.4 | - |
| BINGO | ResNet-34 | 200 | - | 66.1 | Distilled from ResNet-50 |
| TinyMIM | ViT-S | 300 | 83.0 | - | Distilled from Vit-B |
| TinyMIM | ViT-B | 300 | 85.0 | - | Distilled from Vit-L |

Table 26: ImageNet-1k linear probing and fine-tuning benchmarks for **distillation**-based SSL frameworks.

| SSL framework | Backbone network | SSL epochs | Fine-tuning accuracy | Linear probing accuracy | Additional notes |
|---|---|---|---|---|---|
| WMSE | ResNet-50 | 100 | - | 69.4 | $d = 4$, corresponding to 6 positive pairs |
| WMSE | ResNet-50 | 400 | - | 72.5 | $d = 4$, corresponding to 6 positive pairs |
| Barlow Twins | ResNet-50 | 1000 | - | 73.2 | - |
| VicReg | ResNet-50 | 1000 | - | 73.2 | - |
| TWIST | ResNet-50 | 800 | - | 75.5 | Multi-crop |
| TWIST | ViT-B/16 | 300 | 82.8 | 78.4 | - |
| TLDR | ViT-S/16 | 100 | - | 74.8 | - |
| ARB | ResNet-50 | 100 | - | 68.2 | - |
| VicRegL | ConvNext-S | 150 | - | 75.9 | - |
| VicRegL | ConvNext-B | 150 | - | 77.1 | - |

Table 27: ImageNet-1k linear probing and fine-tuning benchmarks for **information-maximization**-based SSL frameworks.

| SSL framework | Backbone network | SSL epochs | Fine-tuning accuracy | Linear probing accuracy | Additional notes |
|---|---|---|---|---|---|
| InfoMin | ResNet-50 | 800 | - | 73.0 | - |
| InterCLR | ResNet-50 + NPID-v2 | 1000 | - | 69.6 | - |
| InterCLR | ResNet-50 + BYOL | 1000 | - | 74.5 | - |
| HEXA | ResNet-50 + MoCo-v2 | 800 | 75.7 | 71.7 | - |
| HEXA | ResNet-50 + DeepCluster-v2 | 800 | 78.3 | 75.5 | 8-crops |
| MocHi | ResNet-50 + MoCo-v2 | 1000 | - | 70.6 | - |
| MSF | ResNet-50 + BYOL-asym | 200 | - | 72.4 | Weak/strong variation |
| MSF | ResNet-50 + BYOL-asym | 200 | - | 66.3 | Weak/weak variation |
| ORL | ResNet-50 + BYOL | 800 | - | 60.7 | Pre-train on COCO+ |
| CEB | ResNet-50 + SimCLR | 1000 | - | 71.6 | - |
| CEB | ResNet-50 + BYOL | 1000 | - | 75.6 | - |
| CEB | ResNet-50 (2x) + SimCLR | 1000 | - | 75.0 | - |
| CEB | ResNet-50 (2x) + BYOL | 1000 | - | 78.8 | - |
| SEM | ResNet-50 + BYOL | 200 | - | 74.1 | - |
| ENS | ViT-B/16 + DINO | 400 | - | 79.1 | - |
| ENS | ViT-B/16 + MSN | 400 | - | 78.9 | - |
| ENS | ViT-B/8 + DINO | 300 | - | 81.0 | - |
| ENS | ViT-B/8 + MSN | 300 | - | 80.8 | - |
| MRCL | ViT-B + SimCLR | 600 | - | 80.0 | - |
| MRCL | ViT-B + BarTwins | 600 | - | 80.4 | - |
| ARCL | ResNet-50 + MoCo | 900 | - | 70.9 | 3 views |
| MosRep | ResNet-50 + MoCo-v2 | 200 | - | 72.3 | - |
| MosRep | ResNet-50 + BYOL | 200 | - | 76.2 | - |

Table 28: ImageNet-1k linear probing and fine-tuning benchmarks for **enhancements** to discriminative SSL frameworks.

| SSL framework | Backbone network | SSL epochs | Fine-tuning accuracy | Linear probing accuracy | Additional notes |
|---|---|---|---|---|---|
| BiGAN | BB | 400 | - | 56.2 | - |
| BigBiGAN | ResNet-50 | 800 | 76.3 | 55.4 | - |
| BigBiGAN | ResNet-50 (×4) | 800 | 76.6 | 60.8 | - |
| SS-GAN | ResNet | 80 | - | 38 | - |
| Vit-VQGAN | ViT-B | 100 | - | 65.1 | - |
| Vit-VQGAN | ViT-L | 100 | - | 73.2 | - |

Table 29: ImageNet-1k linear probing and fine-tuning benchmarks for **GAN**-based SSL frameworks.

| SSL framework | Backbone network | SSL epochs | Fine tuning accuracy | Linear probing accuracy | Additional notes |
|---|---|---|---|---|---|
| iGPT | GPT-L | $\sim 100$ | 72.6 | 65.2 | - |
| iGPT | GPT-XL | $\sim 100$ | - | 68.7 | - |
| iGPT | GPT-XL | $\sim 100$ | - | 72.0 | Concatenation of five layers |
| BEiT | ViT-B | 800 | 83.2 | 56.7 | - |
| BEiT | ViT-L | 300 | 85.2 | 73.5 | - |
| BEiT | ViT-H | 300 | 85.1 | - | - |
| MAE | ViT-B | 1600 | 83.6 | 68.0 | - |
| MAE | ViT-L | 1600 | 85.9 | 75.8 | - |
| MAE | ViT-H | 1600 | 87.8 | 76.6 | - |
| iBOT | ViT-S | 3200 | 82.3 | 77.9 | - |
| iBOT | ViT-B | 1600 | 84.0 | 79.5 | - |
| iBOT | ViT-L | 1000 | 84.8 | 81.0 | - |
| SimMIM | ViT-B | 800 | 83.8 | 56.7 | - |
| PeCO | ViT-B | 800 | 84.5 | - | - |
| PeCO | ViT-L | 800 | 86.5 | - | - |
| PeCO | ViT-H | 800 | 88.3 | - | - |
| MaskFeat | ViT-L | 1600 | 84.0 | 67.7 | |
| data2vec | ViT-B | 800 | 84.2 | - | - |
| data2vec | ViT-L | 1600 | 86.6 | - | - |
| CAE | ViT-S | 300 | 82.0 | 51.8 | - |
| CAE | ViT-B | 1600 | 83.9 | 70.4 | - |
| CAE | ViT-L | 1600 | 86.3 | 78.1 | - |
| CIM | ViT-S | 300 | 81.6 | - | - |
| CIM | ViT-B | 300 | 83.3 | - | - |
| CIM | ResNet-50 | 300 | 80.5 | - | FT 300 epochs |
| MCMAE | ConViT-B | 1600 | 85.0 | 70.9 | - |
| ConMIM | ViT-S | 800 | 83.9 | - | 384 x 384 images |
| ConMIM | ViT-B | 800 | 85.3 | - | 384 x 384 images |
| ConMIM | ViT-L | 1600 | 86.5 | - | 384 x 384 images |
| CMAE | ConViT-B | 1600 | 85.3 | 73.9 | - |
| SdAE | ViT-B | 300 | 84.1 | 64.9 | |
| MILAN | ViT-B | 400 | 85.4 | 79.9 | - |
| MILAN | ViT-L | 400 | 87.8 | 84.3 | - |
| BEiT-v2 | ViT-B | 300 | 85.0 | 80.1 | - |
| BEiT-v2 | ViT-L | 1600 | 87.3 | - | - |
| BEiT-v3 | BB | N/A | 89.6 | - | Uses IN-21k |
| CAE-v2 | ViT-S | 300 | 83.1 | 77.5 | - |
| CAE-v2 | ViT-B | 300 | 85.3 | 80.6 | - |
| CAE-v2 | ViT-L | 300 | 86.7 | 81.7 | - |
| CAN | ViT-B | 1600 | 83.6 | 74.8 | - |
| CAN | ViT-L | 800 | 84.7 | 76.2 | - |
| PCAE | ViT-S | 300 | 81.9 | - | - |
| PCAE | ViT-B | 300 | 83.6 | - | - |
| PCAE | ViT-B | 800 | 83.9 | - | - |
| SparK | ConvX-S | 1600 | 84.1 | - | - |
| SparK | ConvX-B | 1600 | 84.8 | 54.7 | - |
| MRMAE | ConViT-B | 400 | 85.8 | - | - |

Table 30: ImageNet-1k linear probing and fine-tuning benchmarks for **MIM**-based generative SSL frameworks. "SSL epochs" denotes the number of epochs for the SSL training.

# E  COCO benchmarks

| SSL framework | Backbone network | $AP^b$ performance | $AP^m$ performance | Additional notes |
|---|---|---|---|---|
| Swav | ResNet-50 | 41.6 | 37.8 | - |
| Coke | ResNet-50 | 40.9 | 37.2 | - |
| Self-Classifier | ResNet-50 | 41.5 | 36.1 | - |
| InstDist | ResNet-50 | 37.4 | 34.1 | - |
| MoCo | ResNet-50 | 40.9 | 35.5 | - |
| PIRL | ResNet-50 | 38.5 | 34.0 | - |
| SimCLR | ResNet-50 | 39.6 | 34.6 | - |
| MoCo-v2 | ResNet-50 | 39.8 | 36.1 | - |
| DenseCL | ResNet-50 | 39.6 | 35.7 | - |
| PixPro | ResNet-50 | 41.4 | 40.5 | |
| CLSA | ResNet-50 | 42.3 | 24.4 | - |
| MoCo-v3 | ViT-B | 45.5 | 40.5 | - |
| Truncated Triplet | ResNet-50 | 41.7 | 36.2 | - |
| MoBY | Swin-T | 48.1 | 41.7 | - |
| UniVIP | ResNet-50 | 42.2 | 38.2 | - |
| Mugs | ViT-S | 49.8 | 43.0 | - |
| SMoG | ResNet-50 | 40.1 | 36.9 | - |
| SiameseIM | ViT-B | 52.1 | 46.2 | - |
| SimSiam | ResNet-50 | 39.2 | 34.4 | |
| SEED | ResNet-18 | 35.3 | 31.1 | Distilled from ResNet-50 |
| DisCo | ResNet-34 | 40.0 | 34.9 | Distilled from ResNet-50 |
| DINO | ViT-B | 46.8 | 41.5 | - |
| EsVit | Swin-S | 46.2 | 41.6 | - |
| BINGO | ResNet-18 | 34.9 | 31.9 | - |
| Barlow Twins | ResNet-50 | 40.0 | 36.7 | - |
| VicReg | ResNet-50 | 39.4 | 36.4 | - |
| InfoMin | ResNet-50 | 40.4 | 38.8 | - |
| MocHi | ResNet-50 + MoCo-v2 | 39.4 | 34.5 | - |
| ReSim | ResNet-50 + MoCo | 41.9 | 37.9 | - |
| ORL | ResNet-50 + BYOL | 40.6 | 36.7 | Pre-trains on COCO+ |
| MRCL | ViT-B + BarlowTwins | 53.3 | 46.6 | - |
| MRCL | ViT-B + SimCLR | 53.7 | 46.9 | - |
| MosRep | ResNet-50 + BYOL | 41.1 | 37.2 | - |
| MosRep | ResNet-50 + Moco-v2 | 40.6 | 36.6 | - |
| data2vec | ViT-B | 41.1 | 37.0 | - |
| MAE | ViT-B | 48.4 | 42.6 | - |
| iBOT | ViT-S/16 | 49.4 | 42.6 | - |
| SimMIM | ViT-B/16 | 48.7 | 43.6 | - |
| PeCO | ViT-B | 43.9 | 39.8 | - |
| CAE | ViT-B | 50.0 | 44.0 | - |
| MCMAE | ConViT-B | 53.2 | 47.1 | - |
| ConMIM | ViT-B/16 | 48.7 | 43.6 | - |
| CMAE | ViT-B | 52.9 | 47.0 | - |
| SdAE | ViT-B | 48.9 | 43.0 | - |
| MILAN | ViT-B | 52.6 | 45.5 | - |
| MILAN | ViT-L | 55.9 | 48.2 | - |
| BEiT-v3 | - | 63.7 | 54.8 | Uses extra Object365 dataset |
| MRCL | SimCLR | 53.7 | 46.9 | - |
| CAE-v2 | ViT-B | 52.0 | 44.9 | - |
| PCAE | ViT-B | 48.8 | 43.1 | - |
| SparK | ConvX-B | 51.2 | 45.1 | - |
| MRMAE | ConViT-B | 53.4 | 46.9 | - |

Table 31: Downstream transferability results on COCO dataset.

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
