# OpenReview forum: "Know Your Self-supervised Learning: A Survey on Image-based Generative and Discriminative Training"
_TMLR — Accepted by TMLR_

### Review · Reviewer_euRx · 2023-03-04

**Summary Of Contributions:**

This survey paper reviews a number of general-purpose frameworks for image-based self-supervised learning (SSL) that fall into the category of discriminative training. This includes clustering, contrastive learning, distillation, and information-maximization. 50+ SSL techniques are covered.

**Audience:**

No

**Broader Impact Concerns:**

Probably add broad discussions with other related topics, and their shared and unique impact concerns, such as compute cost.

**Claims And Evidence:**

No

**Requested Changes:**

My major concern is how to attract more attention for this survey (in another word, the topic of discriminative SSL), based on the up-to-date literature and the community interests. Some suggestions:

- Try to make more concrete connections with the current community interests, such as (1) generative methods for representation learning & content creation, (2) the possible integration with vision-language pre-training, which is arguably the most empirically effective in visual representation learning for recognition problems.

- Think about why discriminative SSL is not so popular. One reason could be the evaluation metrics. Sticking with the so-called standard metric on "linear probing on ImageNet-1K" does not really present the full power of discriminative SSL methods. Other tasks in computer vision that are aligned with the pre-training objectives could be better downstream evaluation scenarios.

**Strengths And Weaknesses:**


## Strengths
- The important concepts related to SSL are nicely illustrated in Figure 3, with description in Section 3.
- It is good to see the one sentence summary of each of the 50+ SSL techniques.

## Weaknesses
- The topic of discriminative SSL is a bit outdated, due to the fast research pace nowadays. We may use Table 8 as an evidence (the authors summarize the publication information as well as implementation details for the SSL frameworks covered in this survey), the number of papers in 2022 is significantly smaller than 2020 and 2021. In Figure 1, though SSL is shown an increasingly popular topic for the all areas according to Google Trends does not necessarily mean a similar trend for SSL in computer vision, not to mention the discriminative model family in SSL in computer vision. For the information of authors, the community seems more interested in generative SSL paradigm like MAE & BEIT, and vision-language pre-training for general and transferable visual representation learning.

- There are not much new insights and discoveries in this survey, including the future directions section. Though I appreciate the hard work to compile the list and provide the summary, I am afraid that the survey would encourage much more interests in discriminative SSL. The current form of this survey can fit the format of a GitHub readme to maintain the update-to-date papers on this topic.

- Some related papers on discriminative SSL are missing. One simple way to add more related papers is to check out the leaderboard "Self-Supervised Image Classification on ImageNet" on the paper-with-code website: https://paperswithcode.com/sota/self-supervised-image-classification-on
As a survey paper, please at lease add all the methods from this leaderboard, which is crowdsourced by the most related community.

---

> ### Author Response · Authors · 2023-03-15
> **Author response to euRx**
>
> We would like to thank the reviewer for evaluating our manuscript with great effort and sharing our enthusiasm for SSL. We are pleased to hear that the reviewer found the work on discriminative SSL informative, which was our primary objective: engaging interested readers who are inexperienced with discriminative SSL. However, as pointed out by the reviewer _(as well as reviewers xZKh and UR1h)_, our focus on discriminative training and the absence of generative frameworks resulted in a major shortcoming for readers, which we will rectify in the revised manuscript. In what follows, we will address the reviewer's concerns.
>
> **Generative SSL:**
>
> The decision to exclude generative SSL frameworks was a deliberate one on our part, and it was also reflected in the manuscript's title. However, given the reviewer's comment, we understand the relevance of generative SSL to the survey, and we will introduce generative methods as a subsection in Section 5 in a similar fashion to other self-supervision techniques (e.g., contrastive).
>
> **Shortcomings of discriminative SSL and evaluation metrics:**
>
> We agree with the reviewer that judging SSL frameworks solely based on a single benchmark (linear evaluation on ImageNet) is not fair. However, it has been the only common evaluation method for vision SSL frameworks (including generative frameworks). Our goal in using it was to report to readers the frameworks' performance on a common dataset (ImageNet) using a common evaluation method (linear probing). That being said, we share the reviewer's concern in discussing the limitations of discriminative SSL and evaluation methods. We will expand upon those topics and include a number of benchmarks for downstream transfer evaluations.
>
> **Missing frameworks:**
>
> Prompted by the reviewer's comment, we analyzed the link shared by the reviewer and found the majority of missing frameworks to be generative, which we initially did not include in the manuscript. In line with our comment above, in the revised manuscript, we will also cover those generative frameworks.
>
> **Proposed changes:**
>
> In summary, we propose making the following adjustments:
>
> - Introduce generative self-supervised learning (SSL) as a new subsection within Section 4, covering the latest advancements in the field, and update the title of the paper accordingly.
> - Increase the depth of content by discussing the limitations and shortcomings of discriminative SSL frameworks.
> - Expand Section 5 by discussing the current limitations of the evaluation methods (e.g., linear and KNN) and datasets (e.g., ImageNet) used in SSL research.
>
> In addition to the above changes, we will make the following major additions:
>
> - _(cf. feedback of M91J)_ Create a detailed table describing the datasets used for downstream transfer evaluations in SSL frameworks and describe loss functions of SSL frameworks using the notation introduced in Section 3.
>
> With the proposed changes, we hope to address the concerns raised by the reviewer regarding the manuscript's limitations. We are eager to receive feedback from the reviewer and will work diligently to incorporate any necessary revisions as soon as possible, with the aim of submitting the revised version by the first week of April.

---

### Review · Reviewer_M91J · 2023-03-10

**Summary Of Contributions:**

This paper provides a survey of research in image-based discriminative self-supervised learning (SSL). It first provides some historical context and technical background, and then summarizes prior work falling under one of the following SSL learning frameworks: clustering, distillation, contrastive, and information-maximization. It also includes additional resources including links to code implementations and ImageNet benchmark numbers (in appendix).

**Audience:**

Yes

**Broader Impact Concerns:**

No concerns

**Claims And Evidence:**

Yes

**Requested Changes:**

See weaknesses above. I would consider addressing the first two weaknesses as crucial to my recommending acceptance, while addressing the third would strengthen the work but is not crucial.

**Strengths And Weaknesses:**

Strengths

– The paper is very well-written and easy to follow

– The categorization presented is intuitive and fairly comprehensive

– Altogether, the paper is a good one-stop resource for those seeking a summary of recent progress in discriminative SSL on images

Weaknesses

– The paper conceptually explains the different SSL frameworks well but does not provide sufficient formalism, apart from describing the InfoNCE loss, despite introducing considerable notation. I think adding equations describing the generalized loss objective of each framework would help ground many of the concepts that are described much better, even if they may not apply to every method listed within that framework.

– The paper focuses provides a good conceptual comparison of recent SSL approaches but does not provide an adequate head-to-head comparison of their performance, beyond including a table of results on ImageNet. I think additionally compiling results on (atleast a subset of) transfer performance of different approaches, performance v/s model capacity, training time, compute requirements etc. in the appendix would strengthen the paper. Additionally, it would be good to include atleast a brief description of the main takeaways in the main paper, ideally so a reader might be able to judge which method might be suitable for their application of interest.

– Adding "pseudocode" for the different SSL frameworks would be a very nice addition.

---

> ### Author Response · Authors · 2023-03-15
> **Author response to M91J**
>
> We would like to thank the reviewer for spending invaluable time to review our work, identifying its weaknesses and raising a number of valid concerns. We would also like to express our gratitude to the reviewer for evaluating our manuscript from the eyes of a researcher that is outside the field of discriminative SSL, which was the target audience for our work. In what follows, we will try to address the concerns of the reviewer.
>
> **Detailing frameworks with formality:**
>
> We agree with the reviewer that in its current form, the formal concepts  introduced in Section 3 are not utilized well. It is also true that apart from InfoNCE loss, we do not discuss loss functions that are employed in frameworks. We believe the concerns raised by the reviewer on this topic are valid and quite relevant to the target audience of the work.
>
> **Comparison of frameworks and additional information:**
>
> Once again, the reviewer has rightly pointed out that the current version of the manuscript only includes ImageNet evaluations for multiple frameworks, which may seem insufficient. We chose ImageNet evaluations since they are the most commonly used benchmarks across many frameworks. Yet, even for ImageNet, there are a number of frameworks that do not come with benchmarks.
>
> We also agree with the reviewer that it would be ideal to include other information such as performance versus model capacity, training time, performance on downstream datasets, performance on different tasks (such as segmentation and localization), and semi-supervised evaluations with simulated label shortage. Unfortunately, many papers do not include commonly comparable experiments apart from the ImageNet evaluation. In fact, even for SSL frameworks that come from the same research group, downstream evaluations vary. Consequently, while it would be ideal to report the information above, in practice, it is challenging. Nevertheless, we believe that we can address the reviewer's concerns and have proposed a detailed plan below.
>
> **Pseudocodes for frameworks and increasing the glance value:**
>
> We believe Including pseudocodes for each framework would increase the size of the paper considerably and reduce its usefulness. However, to illustrate key features of commonly used SSL frameworks, we will include pseudo-code for a select number of different frameworks (e.g. contrastive, distillation).
>
> **Proposed changes:**
>
> - Given the feedback from the reviewer, we propose the following changes:
>
> - We will introduce commonly employed loss functions used for individual frameworks using the notation introduced in Section 3. This will likely result in a section in the appendix that is dedicated to loss functions.
>
> - In Table 8 in the appendix of our work, we note whether or not a downstream evaluation is performed for the covered frameworks. We propose to remove that column and create an entirely new table that details the datasets used for downstream transfer evaluations as well as the types of problems tackled (i.e., classification, segmentation, localization). With this addition, readers can quickly identify frameworks that experiment with downstream transfer datasets of their interest.
>
> - We believe that adding benchmark information directly to the aforementioned table would make it convoluted due to differing benchmarks being employed for each framework. Instead, we propose to identify common benchmarks for select downstream tasks. For example, we can identify the top three datasets used across all frameworks (e.g., CIFAR, COCO, and Pascal VOC) and report benchmarks for frameworks that have this formation.
>
> - Finally, we will introduce minimalistic pseudocodes for categories of self-supervision.
>
> In addition to the changes requested by the reviewer, we will also:
>
> - _(cf. feedbacks of euRx, UR1h, and xZKh)_ Add generative SSL as a subsection in Section 4 and extend the benchmarks as well as other relevant information (e.g., publication metadata, repositories, abbreviations). With this addition, we propose adjusting the title of the work to "Image-based Generative and Discriminative Training" from "Image-based Discriminative Training."
>
> To summarize, given the opportunity, we believe we can address the concerns of the reviewer. For each of the reviewer’s comments, we detailed our proposals above. If the proposed changes are not in line with the perspective of the reviewer, we would like to engage in a discussion for further adjustments. Given the size of adjustments, we are planning to submit the revised version within the first week of April and hope that this timeline is suitable for the reviewer.

---

### Review · Reviewer_UR1h · 2023-03-14

**Summary Of Contributions:**

Summary: This paper discusses how supervised learning has improved computer vision for images, but the rate of improvement has slowed down. In contrast, self-supervised learning (SSL) has seen great success in natural language processing (NLP) in recent years. This has led to the adoption of SSL methods, such as clustering, contrastive learning, distillation, and information-maximization, in the area of computer vision. The paper reviews research on image-based SSL, covering best practices, software packages, pretext tasks, and techniques commonly used in discriminative SSL. The paper also outlines relevant research directions for those interested in contributing to image-focused SSL.



**Audience:**

Yes

**Broader Impact Concerns:**

No concerns

**Claims And Evidence:**

No

**Requested Changes:**

Adding papers in MAE,  MIM and pre-training on multi-object datasets will be most critical to the survey paper.

**Strengths And Weaknesses:**

Strengths:
1) They have a comprehensive survey of image-based instance discrimination methods capturing both classic and state-of-the-art techniques.

Weakness:
1) The main area of research which is missing is Masked-Autoencoders which has come out recently and has shown great effectiveness.
2) The second area of research is pre-training on multi-object images COCO[1] , OpenImages[2]. Citing all the relevant works in these areas will be important.
3) Methods that have dealt with scaling SSL tasks should also be discussed in the survey.
4) Masked image modelling and it’s related papers should be discussed as well.

References:
[1] Dense-CL CVPR 2021 https://arxiv.org/abs/2011.09157 .
[2] Object-Aware Cropping for Self-Supervised Learning TMLR 2022 https://openreview.net/pdf?id=WXgJN7A69g .

---

> ### Author Response · Authors · 2023-03-15
> **Author response to UR1h**
>
> We would like to thank the reviewer for identifying the shortcomings in our work, which has allowed us to reconsider the focus of the survey to make it more accessible to a larger audience. We are also pleased to hear that the reviewer found the content on discriminative SSL to be comprehensive and informative. In what follows, we will detail our proposals for adjustments according to the reviewer's suggestions.
>
> **Generative SSL:**
>
> The reviewer correctly identified that the survey does not cover generative SSL frameworks _(a point that is also raised by reviewers xZKh and euRx)_. This was a deliberate choice on our part, which is also reflected in the title. However, given the reviewer's comments, we believe that the lack of generative frameworks created a significant gap in the content of the survey, which we will address in the revised manuscript.
>
> **Other advances related to SSL:**
>
> As pointed out by the reviewer, the survey currently does not go into detail for certain subjects that are highly related to SSL, such as multi-image images and scaling SSL models. This was a deliberate choice on our part, given the page limitation as well as the large number of frameworks covered. That being said, in the revised manuscript, we can easily extend discussions in those areas.
>
> **Proposed changes:**
>
> - We will add a subsection for generative SSL in Section 4 and cover SSL frameworks in a similar fashion to others. In doing so, we will cover historic advancements, as well as the most recent developments (MAEs, MIM, and frameworks that make use of MIM). For the frameworks covered, we will also extend the benchmarks, as well as other relevant information (e.g., publication metadata, repositories, abbreviations).
> - In line with the above change, we will adjust the title of the work and replace "Image-based Discriminative Training" with "Image-based Generative and Discriminative Training."
> - We will provide additional details on related areas identified by the reviewer.
>
> Apart from the changes detailed above, we will also make the following major additions:
>
> - _(cf. feedback of M91J)_ We will create a detailed table describing downstream transfer evaluations for SSL frameworks and also detail loss functions employed by SSL frameworks using notation described in Section 3.
>
> If the proposed changes are not in line with the reviewer's perspective, we would like to engage in a discussion for further adjustments. Given the number of revisions, we are planning to submit the revised manuscript within the first week of April and hope that this timeline is suitable for the reviewer.

---

### Review · Reviewer_xZKh · 2023-03-15

**Summary Of Contributions:**

This paper surveys image-based discriminative unsupervised learning. This survey aims to cover image-oriented frameworks for discriminative SSL, some of which may not be covered by the previous survey due to the fast progress in this field. They briefly explain the SSL pretext task, a list of clustering approaches, contrastive learning approaches, distillation, and information maximization.

**Audience:**

Yes

**Broader Impact Concerns:**

No concern.

**Claims And Evidence:**

Yes

**Requested Changes:**

Including MAE papers for more discussion is important. Also, see weaknesses for other changes.

**Strengths And Weaknesses:**

Strength
1. This survey is well organized. Each section is easy to follow.
2. This submission gives a good overview of the recent discriminative self-supervised learning on images.

Weakness
1. One large weakness is that they focus on discriminative self-supervised learning and do not pay much attention to recent methods, such as Masked Autoencoder (MAE). Such approaches are shown to perform well on this task, they need more discussion to clarify the difference from other approaches. Considering that survey in this field already exists, the contribution from this paper can be marginal.
2. Overall, this survey is a bit superficial in that they provide only the list of methods and do not show any comparison in the performance. It is hard to know what approaches are promising for this task. Also, each approach can show strengths and weaknesses depending on the training dataset or downstream task, yet, this paper does not discuss much of it. Although their focus may be to give an overview of this field, I am unsure whether such a survey can be above the bar of acceptance.
3. They lack a discussion on what kind of dataset is used for training and evaluation and the impact of such factors. SSL methods should rely a lot on the dataset they are trained on. ImageNet1K is probably standard in this task, yet increasing the dataset size should further improve the performance, which they do not discuss much.

---

> ### Author Response · Authors · 2023-03-15
> **Author response to xZKh**
>
> We would like to express our gratitude to the reviewer for providing valuable feedback from angles we did not initially consider. We are also pleased to hear that the reviewer finds our work well organized. In what follows we will address the comments of the reviewer:
>
> **Generative SSL:**
>
> As pointed out by the reviewer _(also by reviewers UR1h and euRx)_, our goal was to focus on the developments for discriminative SSL and guide inexperienced researchers that are interested in the field. Nevertheless, this exclusion of generative SSL created a gap in the content of the manuscript which we will address in the revised version.
>
> **Depth of the content:**
>
> Due to the number of frameworks covered as well as page limitations, we had to keep the depth of the manuscript at a maintainable level. In the revised manuscript we will extend discussions on key areas such as: limitations of evaluation metrics, datasets, as well as frameworks.
>
> **Datasets of evaluation:**
>
> Indeed, we have only provided details on ImageNet performance for the frameworks covered since it was the only common dataset of evaluation for many frameworks. Yet, we agree with the reviewer that expanding this information to cover all datasets evaluated in respective papers will be useful for readers.
>
> **Proposed changes:**
>
> - Similar to other self-supervision techniques, we will introduce generative SSL as a subsection into Section 4 and cover advances in this direction extensively. We will also expand benchmarks, abbreviations, as well as metadata information to encompass generative frameworks.
> - We will Increase the depth of content by discussing the limitations and shortcomings of discriminative SSL frameworks as well as evaluation methods used in SSL research.
> - We will create a detailed table describing downstream transfer evaluations for SSL frameworks.
>
> Apart from the changes requested by the reviewer, we will also make the following major additions:
>
> - _(cf. feedback of M91J)_ We will detail the loss functions employed by SSL frameworks with the notation introduced in Section 3 in order to improve formality.
>
> With these changes, we hope to address the reviewer's comments appropriately. Given the magnitude of the changes, we are planning to submit the revised manuscript within the first week of April and hope that this timeline suits the reviewer.

---

### Author Response · Authors · 2023-04-08
**Revision details**

We would like to thank the reviewers once again for their helpful and actionable suggestions. We have modified our manuscript accordingly, and we feel that the revised manuscript has significantly improved as a result. To assist with the reviewing process, we have highlighted all major changes in blue font. We kindly ask reviewers to evaluate our manuscript once again. We hope that this revision meets with their approval and we eagerly look forward to their comments.

In what follows, we provide a brief changelog of the modifications made.

**Generative SSL frameworks:**

In the newly introduced Section 4.3, we covered the most prominent generative SSL frameworks (26) including the most recent approaches that use masked image modelling (MIM). This is a significant addition to the paper and in order to accommodate it, we incorporated the following changes in the rest of the paper::

1- Added MIM and two of its variants as a pretext task in Section 2.

2- Added a basic description of vision transformers in Section 3 (including an illustration in Figure 4).

3- Created two additional visualizations to showcase the overall architectures of discriminative and MIM-based generative frameworks to unfamiliar readers in Figure 5.

4- Added reconstruction targets for MIM-based frameworks (Table 8).

5- Added metadata for generative frameworks (Table 18).

6- Added benchmarks on ImageNet (Table 28 and Table 29)

**Loss functions:**

We briefly detailed a number of popular loss functions used in SSL in the newly introduced Section 3.1, also paying attention to the newly introduced MIM-based generative frameworks.

**Evaluation of SSL frameworks:**

We rewrote Section 5 which contains details on performance evaluation. With that change, we reworked all performance tables in the appendix and condensed their content to bring out the conclusions at a glance. We also included fine tuning results together with linear probing since the former is the preferred method of evaluation for MIM-based frameworks.

**Number of parameters and accuracy:**

We created three new visualizations to show the relation of number of model parameters to accuracy:

1- Figure 6: Linear probing results on ImageNet for discriminative SSL frameworks.

2 & 3- Figure 7: Linear probing and fine-tuning results on ImageNet for MIM-based generative frameworks.

**Dataset usage for SSL frameworks:**

We created an exhaustive list of datasets used in the evaluation of SSL frameworks in the respective papers for discriminative (Table 20 and Table 21) and generative (Table 22) frameworks. We also included the type of downstream evaluations, denoting (*C*) for classification, (*D*) for detection/localization, and (*S*) for segmentation in order to improve the value of the created table.

**Additional downstream performance results:**

We included downstream performance results on COCO for the frameworks that contain this information (Table 30). Note that COCO is the second most-common dataset used across all frameworks after ImageNet.

**Enhancements for existing frameworks:**

We separated model-agnostic enhancements to existing discriminative frameworks from discriminative frameworks and included them in Section 4.2 (with their descriptions given in Table 5 and performance metrics given in Table 27).

**Miscellaneous changes:**

- Reworked appendix in order to improve the readability.
- Tidied the list of abbreviations, grouping together frameworks for each SSL method.
- Rewrote the majority of the conclusions section.
- Included one last discussion topic on the pace of the research and the breadth of the survey.
- Apart from the generative frameworks, included 8 new discriminative frameworks/enhancements.
- Fixed various typos and small errors.
- Moved the appendix before the references in order to facilitate looking up information in the appendix.

In summary, with these changes our revised manuscript now covers all major paradigms in SSL until the present day, and provides an in-depth comparison of SSL frameworks, as evidenced in the reworked and expanded appendix.

---

### Decision · Action_Editors · 2023-04-28

**Recommendation:** Accept as is

**Comment:**

The survey does an excellent job of covering a large amount of literature, without being overly long or verbose. It introduces the core concepts, and provides a concise overview of this broad topic. The paper is well-organized, and uses tables effectively to provide a useful reference/overview highlighting the "unique properties" of the different papers.

There are important topics that are not covered by the survey, such as the role of different pre-training datasets, and transfer of learned features to diverse downstream benchmarks. However, in order to not make the survey too long, I think the current scope is appropriate.

Overall, I recommend acceptance, congratulations to the authors!

**Audience:**

This review covers a popular and rapidly growing field, as evidenced by the large number (100+) methods presented in the paper.

it will be of interest both to those new to the field who are looking for an overview, and those with more familiarity who may want to broaden or contextualize their knowledge in the surrounding literature. There are very many methods covered, so it is unlikely that any reader will not learn of something new from reading this survey.

**Claims And Evidence:**

The paper surveys self-supervised learning for images, focussing on the (large number of) developments in the last ~5 years.

The reviewers and I believe that the survey is comprehensive and provides an excellent overview of the recent literature in both generative and discriminative image-based training. Therefore, the (implicit) claim that the paper adequately surveys the literature is well supported.

---

> ### Author Response · Authors · 2023-05-08
> **Author response to Action Editors and Reviewers**
>
> We would like to express our gratitude to the action editors as well as reviewers for taking the time to re-evaluate our manuscript. We are delighted to hear that the modifications we made have been found satisfactory. We believe that the addition of generative frameworks has tremendously increased the value of our work and enabled us to cover the field of image-based self-supervised learning comprehensively.
>
> In the camera-ready version of the manuscript, we have made the following minor modifications:
>
> - Added metadata information for enhancement frameworks (Table 18).
> - Corrected a number of typos in the manuscript.
> - Revised Figure 1b to improve its readability in black-and-white prints.
> - Corrected a few incorrect references.
> - Fixed notation inconsistencies.